# Multi-step vs. single-step resistance evolution under different drugs, pharmacokinetics, and treatment regimens

**Claudia Igler[1]\*, Jens Rolff[2], Roland Regoes[1]\***

[1]Institute of Integrative Biology, ETH Zurich, Zurich, Switzerland; [2]Evolutionary Biology, Institute for Biology, Freie Universität Berlin, Berlin, Germany

**Abstract** The success of antimicrobial treatment is threatened by the evolution of drug resistance. Population genetic models are an important tool in mitigating that threat. However, most such models consider resistance emergence via a single mutational step. Here, we assembled experimental evidence that drug resistance evolution follows two patterns: (i) a single mutation, which provides a large resistance benefit, or (ii) multiple mutations, each conferring a small benefit, which combine to yield high-level resistance. Using stochastic modeling, we then investigated the consequences of these two patterns for treatment failure and population diversity under various treatments. We find that resistance evolution is substantially limited if more than two mutations are required and that the extent of this limitation depends on the combination of drug type and pharmacokinetic profile. Further, if multiple mutations are necessary, adaptive treatment, which only suppresses the bacterial population, delays treatment failure due to resistance for a longer time than aggressive treatment, which aims at eradication.

**\*For correspondence:**
claudia.igler@env.ethz.ch (CI);
roland.regoes@env.ethz.ch (RR)

**Competing interests:** The authors declare that no competing interests exist.

## Introduction

The rapid rise and spread of antimicrobial resistance severely curb the efficacy of drug treatments against pathogen infections. Treatment strategies are designed to maximize efficacy and limit toxicity, but their long-term applicability depends on the risk of resistance evolution (*Nielsen and Friberg, 2013*; *Foo and Michor, 2009*; *Bonhoeffer et al., 1997*). This highlights the importance of careful consideration of drug type, dose, and duration to guarantee the desired patient outcome whilst also reducing the risk of resistance evolution (*Nielsen and Friberg, 2013*; *Drusano, 2004*). In order to prevent drug resistance and preserve drug efficacy, treatment strategies should also be guided by an understanding of resistance evolution and the ability to assess its risk (*Andersson et al., 2020*; *Read et al., 2011*) – a task that is substantially facilitated through mathematical modeling (*Nielsen and Friberg, 2013*; *Andersson et al., 2020*; *Read et al., 2011*; *Clarelli et al., 2020*).

The main class of models used to predict drug action and treatment outcome are pharmacokinetic and pharmacodynamic (PKPD) models (*Clarelli et al., 2020*; *Drlica, 2003*; *Abel Zur Wiesch et al., 2017*; *Chakrabarti and Michor, 2017*), which describe the change in drug concentration over time (PKs) and the corresponding effect on a pathogen population (PDs). PKPD approaches are most commonly employed to study the efficacy of treatment without considering the possibility of resistance evolution, but coupled with bacterial population models, they can be used to investigate drug resistance evolution over time (*Yu et al., 2018*).

One severely understudied aspect in such approaches is that there are two fundamentally different patterns of de novo antibiotic (AB) resistance evolution: (i) 'single-step' resistance: a single

**eLife digest** The rise in antibiotic resistance is threatening our ability to treat bacterial infections. Bacteria often evolve resistance by acquiring new genetic mutations during the treatment period. Understanding how resistance emerges and spreads through a bacterial population is crucial to prevent antibiotic drugs from failing.

Mathematical models are a useful tool for exploring how bacteria will respond to antibiotics and assessing the risk of resistance. Usually, these models only consider instances where bacteria acquire one genetic mutation that makes them virtually impervious to treatment. But, in nature, this is not the only possibility. Although some mutations do give bacteria a high level of resistance, numerous others only provide small amounts of protection against the drug. If these mutations accumulate in the same bacterial cell, their effects can combine to make the strain highly resistant to treatment. But it was unclear how the emergence of multiple mutations affects the risk of treatment failure and the diversity of the bacterial population.

To answer this question, Igler et al. devised a mathematical model in which each bacterium is able to mutate multiple times during the treatment period. The model revealed that if one mutation provides a high level of resistance on its own, the risk of bacteria surviving treatment is very high. But, if it takes more than two mutations to achieve a high level of resistance, the risk drops to almost nothing.

Igler et al. also found that the chance of bacteria evolving high enough resistance is affected by the type of antibiotics used and how fast the drug decays. With low-level resistance mutations, adapting treatment to maintain an acceptable number of sensitive bacteria as competitors for (a small number of) resistant bacteria was more effective at delaying treatment failure than trying to kill all the bacteria at once.

These findings suggest that adjusting the treatment strategy used for bacterial infections according to the proportion of low- and high-level resistance mutations could slow down the evolution of resistance. To apply these models in the real world, it will be important to measure the level of resistance conferred by single mutations. The type of models used here could also predict the response of other diseases that resist treatment, such as cancer.

mutation provides high drug resistance (*Nielsen and Friberg, 2013*; *Drlica, 2003*; *Yu et al., 2018*); or (ii) 'multi-step' resistance: the accumulation of several mutations of low individual benefit is necessary for high-level resistance (where high resistance here means higher than a given treatment dose). The availability of either pattern to a pathogen population under drug selection will affect the potential for resistance evolution and therefore the evolutionary dynamics in response to various treatment strategies.

We focus on resistance by de novo mutations as long-lasting infections such as those caused by *Pseudomonas aeruginosa* become hard to treat due to resistance evolving via mutations within the host during the course of the treatment (*Oliver et al., 2000*). Another example is tuberculosis (TB), arguably the infectious disease that has caused the highest number of deaths globally (*Castro et al., 2020*). During persistent TB infections, drug resistance evolves by chromosomal mutations while resistance by horizontal gene transfer (HGT) has not been observed (*Castro et al., 2020*). HGT is a common path to resistance in hospital-acquired infections and in cases of shorter treatment durations, as exemplified by *Staphylococcus epidermidis* infections that became resistant by acquiring plasmids carrying genes for linezolid resistance (*Dortet et al., 2018*).

In this study, we will comprehensively study the influence of the mechanistic pattern of resistance evolution itself (namely the benefits and costs of mutations) by considering 'single-step' resistance vs. 'multi-step' resistance. The emergence of mutations and their selection depend on an interplay between various treatment factors like drug type, dose, and treatment duration. These factors have been studied before to various extent in isolation (*Nielsen and Friberg, 2013*; *Drusano, 2004*), although rarely how their interactions shape resistance evolution (*Martinez et al., 2012*; *Olofsson and Cars, 2007*). We will first establish the existence of single-step and multi-step resistance patterns by reviewing evidence in the experimental literature, and then use the obtained

parameter values to inform a stochastic PKPD model of multi-step resistance evolution, which we will explore under various treatment regimens.

We will establish the fundamental differences between evolutionary dynamics emerging from these two patterns in one specific treatment setting, but also explore the impact of various clinically relevant treatment strategies. First, we will compare two types of drugs, ABs and antimicrobial peptides (AMPs). AMPs are key components of innate defenses but also important new antimicrobial drugs, which work by disrupting the bacterial membrane (*Zasloff, 2002*; *Mookherjee et al., 2020*) – as opposed to ABs, which usually target intracellular structures. AMPs have been found previously to significantly reduce the risk of resistance evolution compared to conventional ABs (*Yu et al., 2018*; *Lazzaro et al., 2020*), partly explained by their distinct PDs like higher killing rates (*Yu et al., 2018*). Second, we will consider three different shapes of drug PKs, which are all clinically relevant (*Nielsen and Friberg, 2013*), but have rarely been compared in a systematic manner (*Chakrabarti and Michor, 2017*; *Foo et al., 2012*). These comprise fluctuating drug concentrations, increasing concentrations (which are then maintained at the highest level), and finally constant (which can be achieved in high-dose IV (intravenous) interventions). Third, as a number of recent studies have questioned the practice of 'radical pathogen elimination' (*Read et al., 2011*; *Hansen et al., 2017*; *Hansen et al., 2020*), we will compare aggressive elimination treatment with adaptive suppression – a strategy where the drug concentration is regularly adapted to the pathogen load – in a multi-step mutational framework (*Hansen et al., 2017*; *Gatenby et al., 2009*). Lastly, depending on the drug type, resistance evolution can be shaped either by chromosomal mutations or HGT, or both (*van Hoek et al., 2011*; *Woodford and Ellington, 2007*). Assuming a scenario where both options are available, we will study the relative importance of resistance resulting from de novo mutations as compared to HGT, which plays an important role in AB resistance evolution (*van Hoek et al., 2011*), although likely not as much in AMP resistance (*Kintses et al., 2019*). Taken together, this allows us to obtain an empirically informed modeling framework, which predicts evolutionary dynamics of 'single-step' resistance vs. 'multi-step' resistance in the context of drug type, PKs, and treatment strategies. We show how this framework provides critical insights into drug resistance emergence in clinically relevant treatment settings.

## Results

### Antibiotic resistance evolves via multiple low- or single high-benefit mutation(s)

Experimental studies document single target mutations as well as a sequence of mutational steps to drug resistance evolution in bacterial populations (*Spohn et al., 2019*; *Chevereau et al., 2015*; *Melnyk et al., 2015*; *Lofton et al., 2013*; *Makarova et al., 2018*; *Kubicek-Sutherland et al., 2017*), but no systematic review of these patterns has been conducted so far. Here, we only selected studies that report on both parameters, benefit and costs of resistance (see Materials and methods) (*Spohn et al., 2019*; *Chevereau et al., 2015*; *Melnyk et al., 2015*; *Lofton et al., 2013*; *Makarova et al., 2018*; *Kubicek-Sutherland et al., 2017*), in order to obtain a complete picture of the mutational effects. We define the benefit and cost of a mutation as an increase in the minimum inhibitory concentration (MIC) and as a reduction in growth (in the absence of drug), respectively. Despite differences in study setup and type of resistance mutations, we clearly found a wide range of effects, with a large number of benefits below typical clinical MIC breakpoint values (defining whether a bacterial strain is resistant), which are often 10xMIC or higher (*EUCAST, 2020*; *Table 1*, *Figure 1*) – hence likely necessitating multiple mutations for high resistance. The corresponding fitness costs range from almost none to a 25% reduction of the population growth rate and show a very weak positive correlation ($R^2 = 0.07$, p=0.09) with (log) benefit over all studies taken together (*Figure 1B*, *Figure 1—figure supplement 1*). In general, mutations seem likely to incur more costs than benefits. Notably, our literature search suggests a difference in mutational benefit available for two different antimicrobials: the average benefit of resistance mutations to AMPs is substantially lower than for commonly used ABs (*Table 1*, *Figure 1—figure supplement 1*). In the following, we use the correlation observed with these assembled benefit and cost values to inform a PKPD model that reflects the two patterns of resistance evolution.

**Table 1.** Benefits and costs of drug resistance mutations from experimental studies reported for antibiotics (ABs) and antimicrobial peptides (AMPs), with small mutational benefits (likely giving rise to multi-step resistance patterns) given in blue and large ones (likely giving rise to single-step resistance patterns) in red, assuming a typical clinical drug dose of about 10× minimum inhibitory concentration (MIC) (*Figure 1A, B*).

| Source | Drug type | Organism and evolution environment | Number and type of mutations | Benefit per mutational event | Cost per mutation event | Benefit measurement | Fitness measurement |
|---|---|---|---|---|---|---|---|
| *Spohn et al., 2019* (benefits and costs calculated for individual AB classes are given in *Figure 1—source data 1*) | AMP | *Escherichia coli* K-12 BW25113 populations were evolved in minimal salt (MS) medium over 20 transfers every 72 hr at 30°C (~120 generations) with successively increasing dosages of the antimicrobial | 197 independent mutational events (deletions, insertions, SNPs (Single nucleotide polymorphisms), and intergenic mutations) ~5.2 (±0.8) mutational events/ genome | All MIC samples: 5.1 (±7.2) xMIC ~=1 xMIC/ mutation Only the ones where costs were measured as well: 31.4 (±8.5) xMIC ~=6.0 xMIC/ mutation | 0.2 (±0.16) ~=0.04/ mutation | Serial broth (MS) dilution medium; MIC was defined as OD600 < 0.05 fold-increase compared to ancestor | Continuous monitoring of optical density of liquid cultures (in MS); area under the growth curve from 1 to 24 hr, normalized by the wildtype (WT) |
| | AB | | N.D. Assumed: ~5 | All MIC samples: 72.0 (±2.8) xMIC ~=14.4 xMIC/ mutation Only the ones where costs were measured as well: 120.0 (±2.8) xMIC ~=24.0 xMIC/ mutation | 0.47 (±0.29) ~=0.1/mutation | | |
| *Melnyk et al., 2015* (benefits and costs calculated for individual AB classes are given in *Figure 1—source data 1*) (and references therein: synthesis of 24 studies) | AB | *Borrelia burgdorferi*, *Campylobacter jejuni*, *E. coli*, *Enterococcus faecium*, *Mycobacterium smegmatis*, *Mycobacterium tuberculosis*, *Staphylococcus aureus*, *Streptococcus pneumonia* (all pathogenic) in various environments | Single mutational events (in total 128 mutations) | 96.4 (±19.8) xMIC | 0.13 (±0.24) | MIC fold-increase to ancestor | Competitive fitness (via in vitro growth assays with WT) |
| *Chevereau et al., 2015* | AB | *E. coli* K-12 BW25113 knockout strains (Keio collection) were incubated at 30°C for 20 hr in rich media (LB) with various ABs | Single-gene deletions (3913 mutant strains) ~4 mutations, all types (day 10) | Mecillinam: 1.2 (±1.1) xIC$_{50}$ Trimethoprim: 1.2 (±1) xIC$_{50}$ Ciprofloxacin: 1.3 (±1.2) xIC$_{50}$ ~=18.7 (±1.1) xIC$_{50}$ | 0.13 (±0.07) N.D. | IC$_{50}$ (in LB) | Reduction in growth rate (in LB); calculated from a linear fit of log (OD) in the range 0.022 < OD < 0.22; given relative to WT |
| *Lofton et al., 2013* | AMP | *Salmonella typhimurium* LT2 was passaged daily in rich media (refined LB) at 37°C with successively increasing AMP concentrations for 400–500 generations | SNPs and deletions 2–3 mutational events | LL-37: 1.5–6 xMIC ~=0.5–3 xMIC/ mutation WGH: 12–48 xMIC ~=4–24 xMIC/ mutation CNY100HL: 2–6 xMIC ~=1–3 xMIC/ mutation | 0.076 (±1.1) ~=0.025–0.038/ mutation 0.11 ~=0.037–0.055/ mutation 0.17 ~=0.057–0.085/ mutation | MIC through serial broth (refined LB) dilution | Growth rate from OD600 measurements (in refined LB) in the range 0.02 < OD < 0.2; given relative to WT |

*Table 1 continued on next page*

*Table 1 continued*

| Source | Drug type | Organism and evolution environment | Number and type of mutations | Benefit per mutational event | Cost per mutation event | Benefit measurement | Fitness measurement |
|---|---|---|---|---|---|---|---|
| *Kubicek-Sutherland et al., 2017* | AMP | *S. aureus* (MRSA) WT JE2 (DA28823; clinical isolate) was passaged daily in minimal media (MIEM) at 37°C with successively increasing AMP concentrations | 1–3 (adaptive) amino acid substitutions | LL-37: 6.5 (±6.2) × survival (for this AMP benefit was measured not as fold MIC increase but fold bacterial survival) ~=2.2–6.5 xMIC/ mutation WGH: 13.9 (±1.2) xMIC ~=4.6–14 xMIC/ mutation PR-39: 2 (±1) xMIC ~=0.7–2 xMIC/ mutation | <= 0 (no detectable cost, rather an advantage, was found in the media used in the evolution experiment) | MIC through serial broth (MIEM) dilution, except for LL-37 where kill curves were used (in tryptic soy broth, rich media) | Maximum growth rate in MIEM based on (exponential phase) OD600 measurements; given relative to WT |
| *Makarova et al., 2018* | AMP | *S. aureus* SH1000 and *E. coli* MG1655 were transferred for seven daily passages in rich media (Mueller Hinton Broth [MHB]) at 37°C with successively increasing AMP dosages | All types ~2.4 (±0.9) mutational events | Tenecin 1: 3.1 (±1.1) xMIC of wt ~=1.3 xMIC/ mutation | 0.25 (±0.13) ~=0.1/mutation | MIC through serial broth (MHB) dilution; defined as inhibition of visible growth after 24 hr | Maximum growth rate in MHB based on OD600 measurements |
| | AMP | | | Mean: 4.5 (±5.3) xMIC/mutation | Mean: 0.04 (±0.03)/ mutation | | |
| | AB | | | Mean: 28.0 (±7.1) xMIC/mutation | Mean: 0.10 (±0.07)/ mutation | | |

## The PD model

We mainly investigated the rise of de novo resistance in a clonal, susceptible pathogen population, which is a common starting point for many clinically relevant infections (*Balmer and Tanner, 2011*), by extending a previously described stochastic PKPD model (Materials and methods) (*Yu et al., 2018*). Specifically, we considered not only a single resistance mutation, but the potential emergence of a sequence of mutations, with each mutation conferring a certain (additional) benefit and cost (*Figure 1*). The number of mutations needed for 'full' resistance depends on the applied drug dose, but generally low mutational benefits are more likely to necessitate multi-step resistance evolution. To compare scenarios where a single mutation is sufficient to scenarios where several mutations have to arise in one cell, we ran the simulations over a range of mutational benefits (2–100 xMIC) – and their correlated fitness costs (*Table 1*, *Figure 1*) – in combination with various drug doses (0.5–100 xMIC). Hence, the minimum number of mutations necessary for resistance was predetermined (*Figure 1—figure supplement 2*), and we investigated how this affects the potential for pathogen survival and mutational diversity under various treatment strategies (PKs) and for two different antimicrobials (PDs) as described below. Competition between the various mutant subpopulations was modeled by imposing a carrying capacity for bacterial growth and very low turnover as soon as this capacity is reached.

## Multi-step resistance patterns show lower risk of treatment failure and lower genetic diversity

First, we determined the probability of treatment failure by simulating change of the pathogen population over 200 hr under treatment with drugs (PD) parameters typical for bactericidal ABs (*Yu et al., 2018*; *Supplementary file 1*) being applied once every 24 hr (PK). We assumed that the

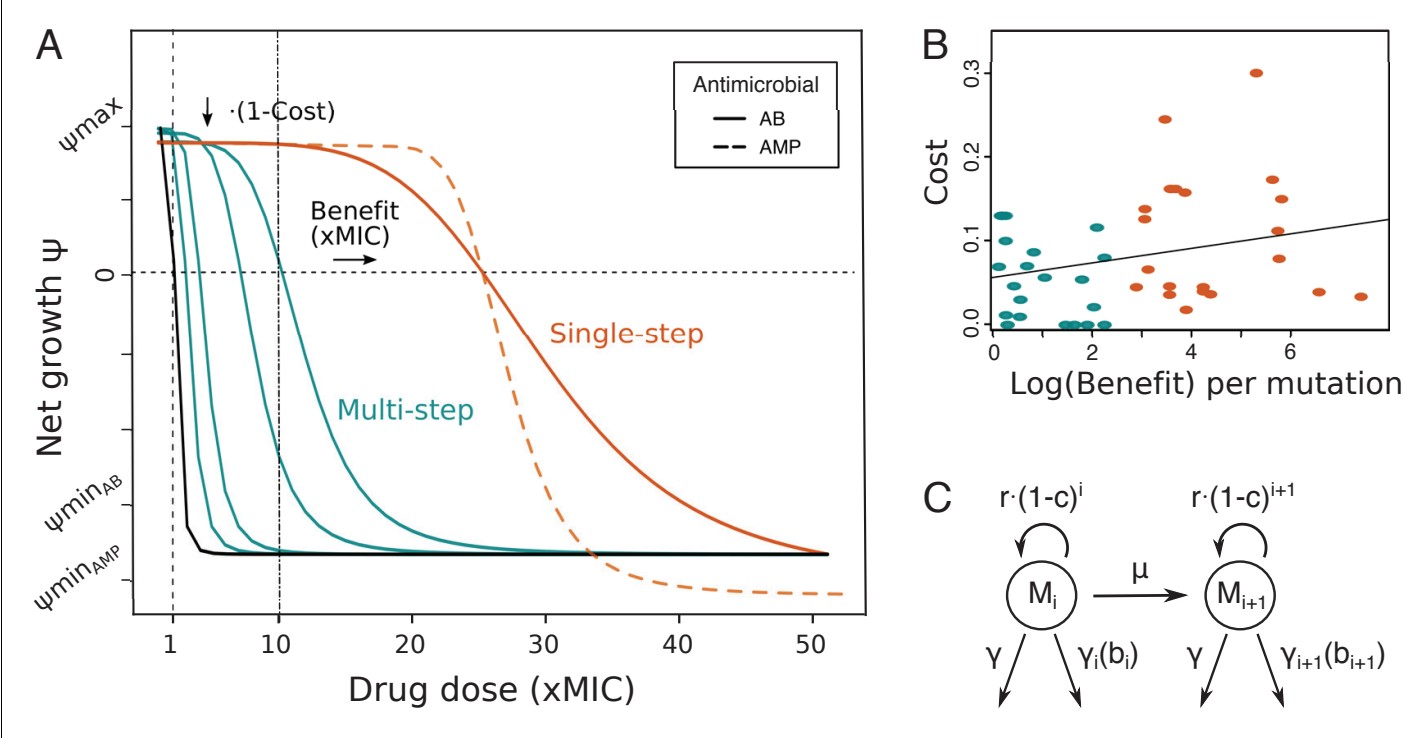

**Figure 1.** Pharmacodynamic (PD) model of single- and multi-step resistance. (**A**) The PD curve relating bacterial net growth $\psi$ (which is between the maximal growth rate $\psi_{max}$ and the maximal killing rate $\psi_{min}$) to antimicrobial drug concentration (given in fold minimum inhibitory concentration (MIC)) illustrating a sensitive wildtype (black) and mutants with either small (blue) or large (orange) MIC increases per mutation (benefit), assuming a typical clinical drug dose of 10xMIC. Characteristic PD curves for antibiotic (AB) (solid lines) and antimicrobial peptide (AMP) (dashed lines) single-step resistance are compared (orange), demonstrating the typically steeper decrease ($\kappa$) and lower $\psi_{min}$ observed with AMPs. (**B**) Shown are costs and benefits from various empirical studies (*Table 1*), each dot representing resistance mutations to a specific AB or AMP class. The cost of a mutation shows a very weak positive correlation with the log (benefit) ($R^2 = 0.07$, $p=0.09$). Blue and orange colors show multi- or single-step resistance benefits given the drug dose in (**A**). (**C**) Schematic of the PD model with several mutated subpopulations ($M_i$), which grow with a cost $r(1-c)^i$, determined by the number of mutations $i$, mutate with rate $\mu$, and die at a constant rate $\gamma$ and a drug-specific rate $\gamma_i(b_i)$, which is dependent on the benefit $b_i$ conferred per mutation.

The online version of this article includes the following source data and figure supplement(s) for figure 1:

**Source data 1.** Empirical data used to obtain *Figure 1B*.

**Figure supplement 1.** Direct comparison of mutational benefits and costs between antimicrobial peptides (AMPs) and antibiotics (ABs) from *Spohn et al., 2019*.

**Figure supplement 2.** Number of mutations necessary for resistance.

pathogen population initially consists of completely susceptible bacteria and defined a treatment as failed if the pathogen population was not eradicated after 200 hr. We found that the probability of treatment failure was always close to 1 for single-step resistance evolution, but decreased rapidly if multiple mutations were required. Notably, already if three mutations were necessary to overcome the applied dose, the probability of pathogen survival approached 0 (*Figure 2A*, *Figure 1—figure supplement 2*). The qualitative picture of these results was not dependent on the specific cost-benefit correlation that we are assuming for most of our simulations (*Figure 2—figure supplement 1*).

One aspect of resistance evolution that is especially important when considering multiple mutations is the mutational diversity that arises in the pathogen population: high genetic diversity (here meaning diversity in the resistance phenotype) increases the probability that some individuals will be able to survive a given environment – such as treatment with other drugs (*Castro et al., 2020*) – and increases the adaptive potential overall (*Van Egeren et al., 2018*). Using the Shannon index to determine the highest mutational diversity obtained in the population over the treatment period, we clearly observed higher diversity with single-step than multi-step resistance evolution (*Figure 2B*), even if we increased the mutation rate proportionally to the number of mutations required

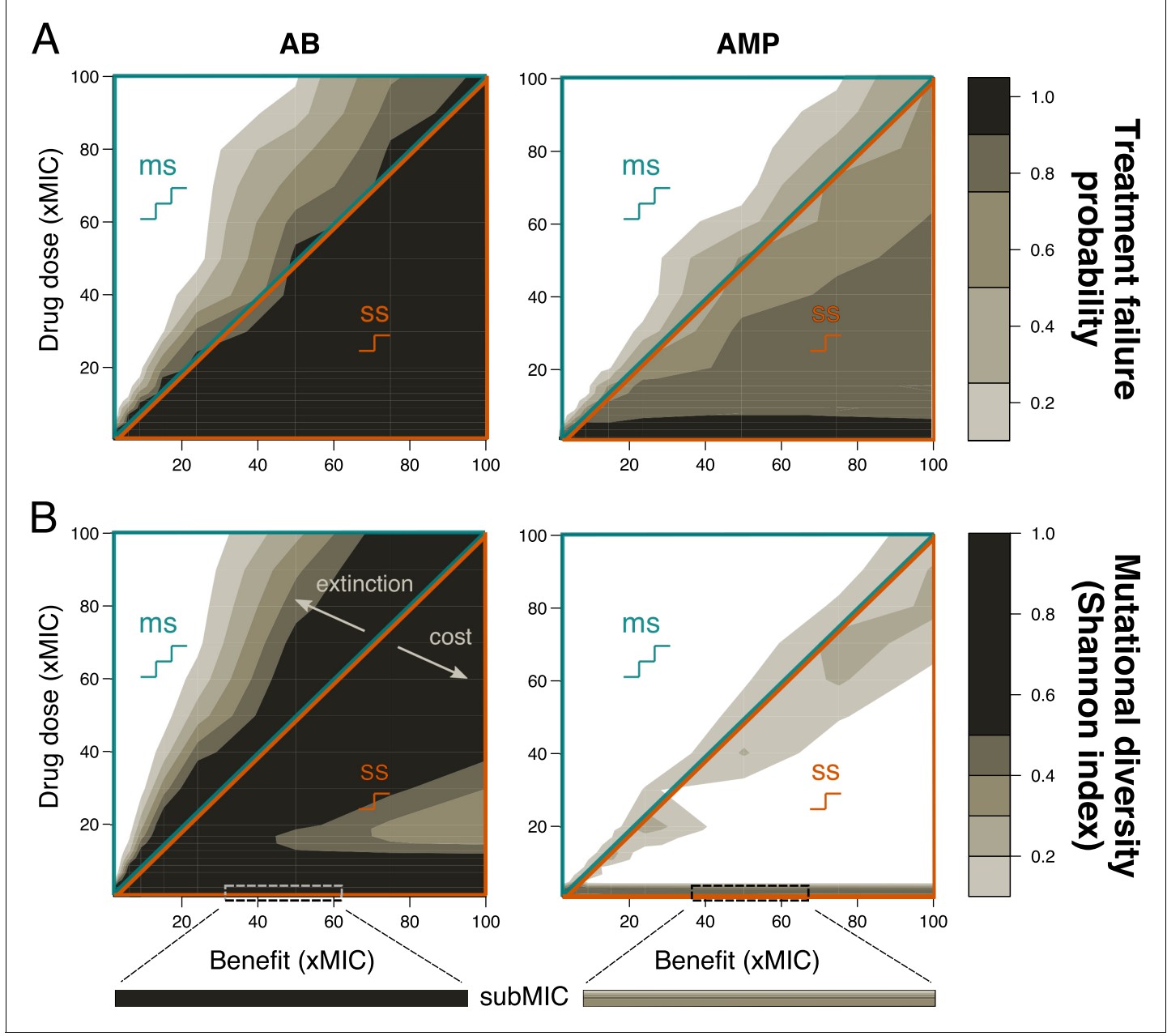

**Figure 2.** Resistance evolution with single- and multi-step patterns for peak pharmacokinetic (PK). (**A**) Treatment failure probability (measured at the end of the treatment period) and (**B**) mutational diversity (calculated over the whole treatment period) are shown for two different antimicrobial classes (antibiotics (ABs), left; antimicrobial peptides (AMPs), right) for different combinations of mutational benefits (xMIC) and drug doses (xMIC). The diagonal line shows where the benefit per mutation is exactly equal to the given drug dose and separates single-step (ss, lower orange triangle), where one mutation gives a benefit higher than the applied dose, from multi-step (ms, upper blue triangle) resistance, where more than one mutation is needed for the accumulated benefit to match the drug dose. The arrows indicate the decrease in diversity either due to increasing extinction (too many mutations are needed for survival) or due to increasing cost per mutation (costs are too high compared to the additional benefit). A representative example of subMIC mutational diversity is shown magnified below the plots in (**B**). MIC: minimum inhibitory concentration.

The online version of this article includes the following figure supplement(s) for figure 2:

**Figure supplement 1.** Resistance evolution with single- and multi-step patterns for peak pharmacokinetic (PK) with steeper correlation between cost and log(benefit).

**Figure supplement 2.** Resistance evolution with increased mutation rates (proportional to the number of mutations required for resistance).

**Figure supplement 3.** Relative population frequencies with horizontal gene transfer (HGT).

**Figure supplement 4.** Treatment failure is similar with and without horizontal gene transfer (HGT).

**Figure supplement 5.** Resistance evolution with random mutational benefit and cost.

*Figure 2 continued on next page*

*Figure 2 continued*

**Figure supplement 6.** Resistance evolution with single-step (ss) and multi-step (ms) patterns for peak pharmacokinetic (PK) starting from neutral mutation heterogeneity.

(*Figure 2—figure supplement 2*). It can be shown analytically that a mutant strain can invade at the mutant-free equilibrium if the death rate of the sensitive strain is higher than the death rate of the mutant, where the mutant death rate is a combination of intrinsic and drug-induced death as well as the mutational cost (Materials and methods). The observed higher diversity with single-step patterns seems counterintuitive as the need for multiple mutations should increase diversity (*Figure 2B*), but can be explained as follows: at high drug doses and low benefits, this effect is due to extinction that effectively reduces genetic diversity, while at low doses and high benefits, high mutational costs inhibit the build-up of diversity. These findings agree with an experimental study showing that resistance alleles with low costs are favored (*Wichelhaus et al., 2002*).

## Consistently lower treatment failure with multi-step resistance for various PKs and PDs

Our results clearly show less resistance if multiple mutations are necessary, but the relative importance of the number of resistance mutations compared to other treatment considerations like the dose-response profile of a drug (PD) (*Yu et al., 2018*; *Spohn et al., 2019*) or the administration mode (PK) required further investigation. Hence, we compared three different PKs: 'peak' (fast absorption and exponential decay), 'ramp' (slow, linear absorption and no decay), and 'constant' (immediate absorption and no decay) (*Figure 3A*). Whereas constant PKs distinctly lowered the probability of treatment failure and the emergence of mutational diversity, peak and ramp PKs showed similar magnitudes of resistance evolution (*Figure 3B, C*, *Figure 3—figure supplements 1* and *2*). However, ramp PKs lead to more than twice the mutational diversity with multi-step resistance patterns (*Figure 3—figure supplement 1*), which suggests that treatment failure and pathogen diversity are connected in a non-trivial manner: while higher mutational diversity increases the risk of resistance evolution, neither its presence nor absence is obviously predictive of the treatment outcome (*Figure 2*, *Figure 3—figure supplements 1* and *2*).

The evolutionary dynamics can also be contrasted for different antimicrobial drugs, AMPs and ABs, by using two different PD parameter sets (Materials and methods, *Figure 1—source data 1*). Briefly, AMP treatments are characterized by higher killing rates, steeper dose-response curves (*Figure 1A*), and lower mutation rates than AB ones (*Yu et al., 2018*). Consistent with previous findings that AMPs lead to a lower risk of resistance evolution and a narrower mutant selection window (MSW) than ABs (*Yu et al., 2018*), treatment failure and mutational diversity was lower for AMPs with peak and constant PK treatments (*Figures 2* and *3*, *Figure 3—figure supplement 2*). Notably, in accordance with empirical studies (*Andersson and Hughes, 2014*), we generally see mutations accumulating at sublethal drug doses, but the maximal diversity is substantially lower in AMP treatments (*Figure 2B*, *Figure 2—figure supplement 1*).

Interestingly, the steeper dose-response curve of AMPs seems to make their resistance dynamics more sensitive to the shape of the PK than those of ABs (*Figures 2* and *3*, *Figure 3—figure supplements 1*, *2,* and *4*): in contrast to the other two PK profiles, ramp PKs lead to a drastic increase in treatment failure with AMPs, especially in multi-step scenarios (*Figure 3*, *Figure 3—figure supplement 1*). Accordingly, for ramp PKs, AMPs did not perform better and under some conditions even worse than ABs (*Figure 3—figure supplement 4*). By varying the ramp duration (or equivalently the rate of drug uptake), we found that there is an intermediate range (48–84 hr), which showed increased treatment failure with AMPs over ABs (*Figure 3—figure supplement 5A*). Paradoxically, while a narrow MSW generally hinders the emergence of numerous mutations in the population, for ramp PKs it can lead to optimal selection conditions for the sequential emergence of increasingly higher resistance mutations due to the strong selection for the next mutation combined with sufficient time for its emergence. Hence, especially the risk of multi-step resistance is increased if AMPs are used with ramp treatments as compared to the other PKs (*Figure 3B, C*). The broader selection windows in the presence of ABs, on the other hand, overlap and resistance mutations are less strongly favored (*Figure 3—figure supplement 5B*). Overall, the number of resistance mutations

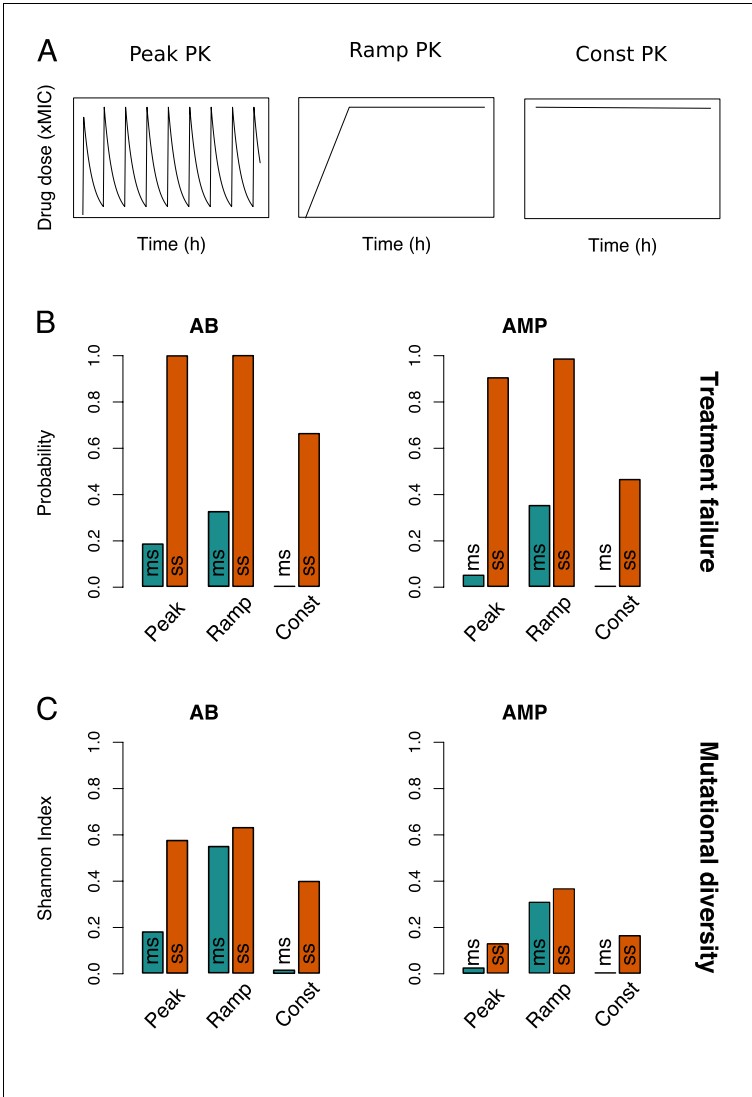

**Figure 3.** Resistance evolution patterns with different pharmacokinetics (PKs). (**A**) The three PKs used in the model are shown over time (in hours) for the same peak drug concentration (xMIC). (**B**) The treatment failure probabilities and (**C**) mutational diversities are given for the three PKs from (**A**) and two antimicrobial drug classes (antibiotics (ABs) and antimicrobial peptides (AMPs)). Blue (orange) bars show averages calculated over the blue (orange) framed triangular areas of multi-step (ms) (single-step (ss)) resistance evolution shown in *Figure 2*, *Figure 3—figure supplements 1* and *2*. MIC: minimum inhibitory concentration.

The online version of this article includes the following figure supplement(s) for figure 3:

**Figure supplement 1.** Resistance evolution with single-step (ss) and multi-step (ms) patterns for ramp pharmacokinetic (PK).

**Figure supplement 2.** Resistance evolution with single-step (ss) and multi-step (ms) patterns for constant pharmacokinetic (PK).

**Figure supplement 3.** Mutational diversity at the end of the treatment period.

**Figure supplement 4.** Comparison of resistance evolution with antibiotic (AB) and antimicrobial peptide (AMP) treatments using peak or ramp pharmacokinetic (PK).

**Figure supplement 5.** Selection coefficient analysis.

**Figure supplement 6.** Resistance evolution patterns with different pharmacokinetics (PKs) and pharmacodynamics (PDs) for bacteriostatic drug action.

**Figure supplement 7.** Resistance evolution patterns with different pharmacokinetics (PKs) for simulations starting from neutral mutation heterogeneity.

was the main determinant of treatment outcome, but we also found a complex dependence on PK and PD characteristics.

This complexity in resistance determinants raises the question in how far the type of drug action influences treatment outcome. Specifically, antimicrobials can have bactericidal action (which we were modeling so far, through a drug-dependent death rate), but they can also act bacteriostatically, that is, decreasing bacterial growth. We would expect bacteriostatic antimicrobials to slow down the rise of mutations in comparison to bactericidal ones as the acquisition of mutations is also coupled to bacterial growth. However, we find that this is only true for ramp and constant PK treatments (*Figure 3—figure supplement 6A*). Peak PKs allow for regrowth of bacterial cells due to drug decay, which increases bacterial survival and treatment failure, especially with multi-step resistance. Interestingly, mutational diversity only increased for AMP treatments (*Figure 3—figure supplement 6B*).

## Multi-step resistance can lower the threshold for adaptive treatment application

The conventional treatment goal is to 'eradicate' the pathogen population, but it has been suggested that under certain conditions 'mitigation' could be a superior strategy (*Hansen et al., 2017*; *Hansen et al., 2020*; *Gatenby et al., 2009*), for example, if it is likely that a resistant subpopulation already exists at the beginning of the treatment. This strategy is called adaptive treatment as drug doses are adapted to keep the sensitive population as big as possible and the total pathogen burden below a given limit. (In practice, this is challenging as it requires monitoring of the pathogen burden and adjusting drug doses accordingly, which is difficult to implement even for measurements of total within-patient loads.) In adaptive treatment, the sensitive population provides a benefit by competitively inhibiting the resistant subpopulation, but also a risk by supplying mutational input (*Figure 4*). This trade-off creates a threshold for the size of the pre-existent resistant subpopulation above which adaptive treatment is more effective than aggressive 'eradication' in containing the infection (*Hansen et al., 2017*).

Previously, the threshold for adaptive treatment was derived in a single-step resistance scenario (*Hansen et al., 2017*). When we incorporated adaptive treatment in our multi-step resistance framework (*Figure 4—figure supplement 1*), we found that the resistant subpopulation threshold above which adaptive treatment is more beneficial can be much lower in the multi-step scenario than in the single-step one (*Figure 4A*). This can be intuitively explained by the fact that all (partially) sensitive bacteria serve as competitors for fully resistant cells, but only the subpopulation one mutation away from being fully resistant constitutes the risk population (*Figure 4B*). Thus, with multi-step resistance there is a smaller population to supply resistant bacteria than with single-step resistance, changing the trade-off towards adaptive treatment. Additionally (in scenarios where adaptive treatment is favorable), the difference between adaptive and aggressive treatment in the duration until treatment failure can be several-fold larger for multi-step than single-step resistance patterns (*Figure 4—figure supplement 2*). Hence, assuming either single- or multi-step evolution could lead to considerably different treatment strategy assessments with regard to treatment failure through resistant pathogens.

## HGT does not change the treatment failure probability

In addition to chromosomal mutations (*Woodford and Ellington, 2007*), antimicrobial resistance can be conferred through HGT (*van Hoek et al., 2011*), which could facilitate resistance in multi-step scenarios. To account for this possibility, we extended the model to allow for acquisition of a gene conferring full resistance, initially only at a low rate from the environment, and then at a density-dependent rate from other cells carrying the HGT gene (for assumption and implementation details, see Materials and methods). The HGT gene always provided immediate resistance to the applied maximal dose, regardless of the benefit or costs conferred by mutations. In order to compare the population dynamics of these two main antimicrobial resistance acquisition mechanisms, we assumed that resistance through mutations or HGT can be acquired independently of each other and that their effects are multiplicative.

Even though HGT carriers dominated the remaining pathogen population at the end of the treatment (*Figure 2—figure supplement 3*), the addition of HGT did not change the probability of treatment failure (*Figure 2—figure supplement 4*). This result holds true as long as the acquisition rate

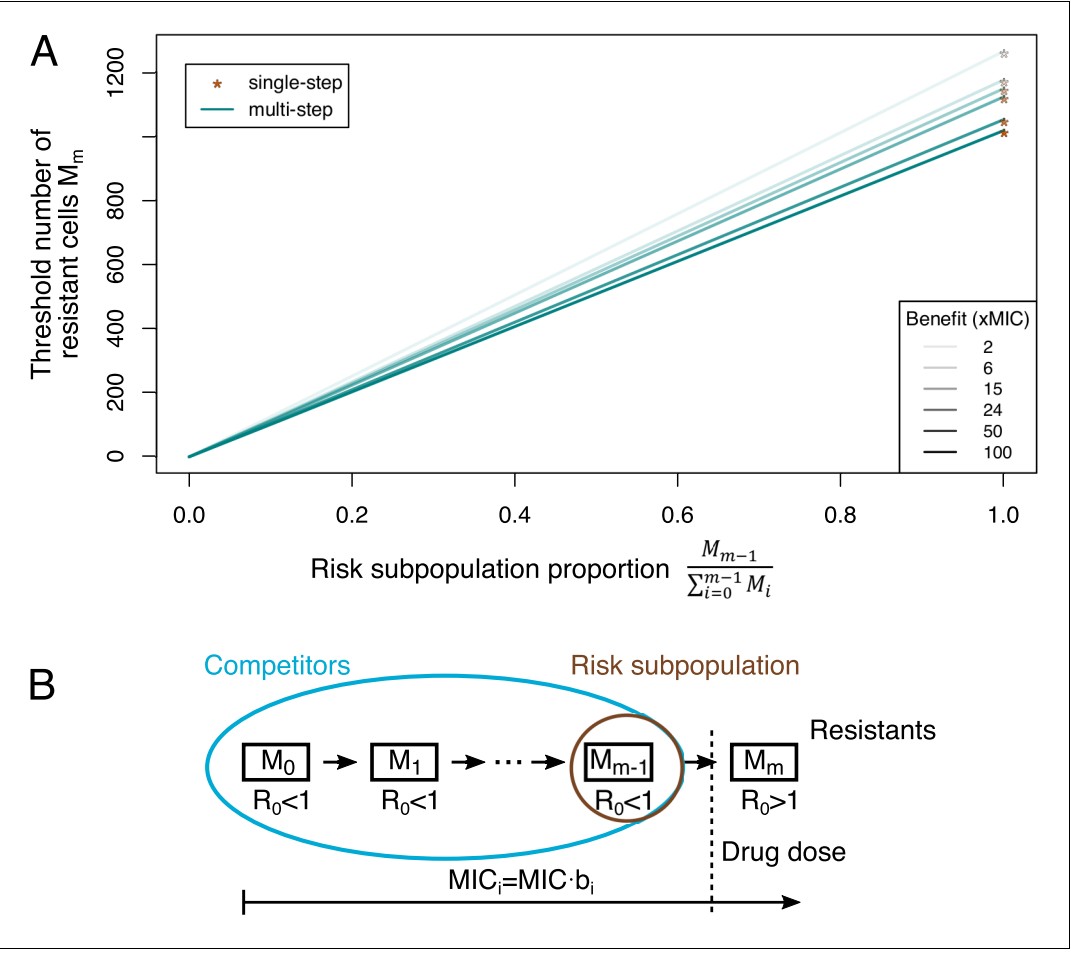

**Figure 4.** Adaptive treatment threshold. (**A**) The dependence of the threshold number of resistant bacteria cells $M_m$ is given for different proportions of the risk population to the whole competitor population for single-step (orange stars), where the risk population is always equal to 1, and multi-step (blue lines) resistance patterns. Different benefits (and correlated costs) per mutation are shown as different color shades. (**B**) The minimum inhibitory concentration (MIC) increases with every mutation (given by $b_i$), but only an MIC above the given drug dose will lead to a reproductive number $R_0 > 1$, that is, growth of the population (resistant cells). All other subpopulations serve as competitors, and the subpopulation one mutation away from resistance is the risk population.

The online version of this article includes the following figure supplement(s) for figure 4:

**Figure supplement 1.** Optimal pharmacokinetic (PK) for adaptive treatment.

**Figure supplement 2.** Differences in time to treatment failure are more pronounced with multi-step patterns.

from the environment is lower than the mutation rate (this constraint is examined further in the Discussion). Consequently, initial rescue of the population is due to mutations – and therefore dependent on the magnitude of the mutational benefit – whereas HGT resistance is acquired later during the infection, after which it spreads rapidly.

## Discussion

In this study, we compared the risk of drug resistance evolution patterns that either feature single resistance mutations with large costs and benefits or multiple steps involving mutations with smaller costs and benefits. We extended this comparison across a wide range of PD and PK profiles, which cover a multitude of antimicrobials and treatment strategies. We first showed that the single- and multi-step resistance patterns are relevant by gathering evidence of multi-step resistance patterns in the experimental literature (*Table 1*, *Figure 1*). While it is intuitive that drug resistance requiring

more than one mutation will arise more slowly, we find that it can be a surprisingly strong inhibitor of resistance evolution and mutational diversity, depending on the drug class and administration route (*Figure 3*). We demonstrated that the number of mutations necessary for resistance strongly affects predictions of treatment outcome and optimality with regard to antimicrobial resistance – in a manner that is robust to variations in mutation rates and in the cost per mutation (*Figure 2—figure supplements 2* and *5*). Experimental support for our simulation results comes from studies reporting that mutational input limited to low benefits (*Drlica, 2003*) leads to decreased drug resistance evolution as compared to systems, in which high-benefit mutations are available (*Allen et al., 2004*). Moreover, limited access to high-benefit mutations seems to curtail MIC increase beyond a certain threshold (*Chevereau et al., 2015*).

The pattern of resistance evolution (single- and multi-step) is likely to be associated with the molecular mechanisms of resistance for a given antimicrobial: as an overall rule, the magnitude of the resistance benefit correlates with the mechanism of resistance, for example, efflux pumps yield low benefits, whereas specific drug target mutations yield high benefits (*Hughes and Andersson, 2017*).

Unfortunately, the specific mutations linked to the benefit and cost of mutations in our literature analysis (*Table 1*) are generally not known. Overall, however, MIC increase was low for drugs, which typically show unspecific resistance mechanisms via two-component systems or lipopolysaccharide modifications – as generally seen for AMP resistance (*Lofton et al., 2013*; *Makarova et al., 2018*; *Kubicek-Sutherland et al., 2017*) – and high for drugs with typical resistance via specific target modifications, as seen for some AB classes (e.g., rifampicin resistance via RNAP subunit mutations) (*Goldstein, 2014*). For most drugs, the prevalent resistance mechanisms are known (*van Hoek et al., 2011*); hence, this information can be used to determine drug and dosing regimens that minimize resistance evolution based on the inferred pattern of resistance evolution (i.e., using the probability that a single- or multi-step pattern is underlying resistance evolution). A recent study also suggests that resistance evolution in biofilms, which are often associated with clinical infections, is prone to occur through unspecific mechanisms, even if specific mechanisms are favored in planktonic cultures (*Santos-Lopez et al., 2019*).

Interestingly, the risk of resistance evolution does not seem to be related to the emerging mutational diversity in the population in a trivial manner as it is either limited by fast extinction or high mutational cost (*Figure 2*, *Figure 3—figure supplements 1* and *2*). Reducing mutational diversity is however a worthwhile goal in its own right as mutational diversity can increase adaptation by fixing more mildly deleterious mutations, which can then act as stepping stones for multi-drug resistance evolution (*Van Egeren et al., 2018*). Further, the diversity arising during the treatment period will help to determine if escalating the drug dose is expected to be beneficial or if, conversely, it would be detrimental because higher-resistance mutations are already present in the population and would be selected. The strength of this selection is determined by the MSW of the antimicrobial. Hence, the number of resistance mutations emerging over the treatment can be useful in estimating the width of the MSW – even though the diversity remaining at the end of the treatment will likely be lower (*Figure 3—figure supplement 3*).

We find that mutational diversity arises from a combination of selection pressure, bacterial growth, and fitness costs and cannot be predicted from the mutational benefit or the probability of treatment failure alone. Diversity is also shaped in unexpected ways by interactions between the drug type and drug concentration changes, making drug choice not only dependent on the PD characteristics, but also the specific drug PK in the target body compartment. Notably, this can lead to more favorable assessment of a specific drug application mode for one type of drug (e.g., AMPs for bolus drug application), but a different mode for another drug (e.g., ABs for drug infusions). While we mostly focused on the action of bactericidal drugs in this study, we note that purely bacteriostatic effects can lead to different trends for PK and PD influence on treatment failure and mutational diversity, for example, making peak PKs the least favorable drug administration route (*Figure 3—figure supplement 6*). The unexpected complexity in predicting which treatment strategies will minimize resistance evolution highlights the need of critically evaluating assumptions such as single-step resistance made in current PKPD models. The role of specific drug characteristics in resistance evolution is exemplified by the steepness of the PD curve, κ. By analyzing the selection coefficients for various treatments, we find that κ governs not only the size of the MSW (*Yu et al., 2018*; *Chevereau et al., 2015*), but generally shapes the selection pressure for resistance evolution in a

qualitative manner. $\psi_{min}$, the minimal bacterial growth rate, on the other hand, leads to substantial quantitative changes in selection pressure, meaning that $\kappa$ and $\psi_{min}$ shape the form and strength of drug selection independently (*Figure 3—figure supplement 5C*). Ultimately, the interactions between PD and PK characteristics give rise to complex, and dynamic, fitness landscapes that are navigated by mutations of various benefit and cost sizes.

Interestingly, AMP-like drugs show considerably more resistance evolution with ramp PKs than in the other PK scenarios. This is noteworthy as natural AMP expression patterns in the producing organisms resemble ramp PKs (*Johnston et al., 2014*; *Haine et al., 2008*). This finding could suggest another reason why natural AMP production in cocktails is favorable (*Zanchi et al., 2017*) as AMP cocktails will limit the selection pressure and potential for resistance evolution to individual components. Intuitively, we would expect that a gradual increase in drug concentration would facilitate the rise of multiple mutations and indeed we find that ramp PKs lead to the highest probability of treatment failure and mutational diversity (*Figure 3*). However, a high probability of treatment failure is still mostly observed with high mutational benefits (*Figure 3—figure supplement 1*), that is, limited with the small-benefit mutations likely associated with multi-step resistance (*Jochumsen et al., 2016*). For clinical settings, our simulations caution that attention should be paid to the drug application mode when using AMPs. AMP-like colistin and daptomycin, for example, are typically applied as (short) IV treatments (*Liu et al., 2011*; *Tsuji et al., 2019*), which resemble peak PKs, and they are still active as last-resort drugs for multi-drug-resistant bacterial pathogens (*Liu et al., 2011*; *Tsuji et al., 2019*). Overall, our results agree with *Yu et al., 2018* in that AMP treatment lowers resistance evolution and mutational diversity. This is particularly notable as multi-step patterns seem to be the common mechanism by which AMP resistance evolves (*Table 1*; *Spohn et al., 2019*; *Jochumsen et al., 2016*; *Joo et al., 2016*) – thereby suggesting another advantage over ABs, for which single- and multi-step evolution is common (*Drlica, 2003*; *Weinreich et al., 2006*; *Wistrand-Yuen et al., 2018*; *Marcusson et al., 2009*; *Jin and Gross, 1988*).

Unfortunately, distributions of mutational effects have rarely been characterized experimentally for drug resistance, and even then only for a single mutational step (*Chevereau et al., 2015*). We show, however, that this information is crucial as input for PKPD models to accurately predict resistance evolution and population diversity in response to drug treatment. Even between mutations involved in multi-step resistance to a single drug, benefit and costs of individual mutations are likely to vary (*Figure 1B*). In addition, epistasis in benefit and or cost magnitude can facilitate or preclude certain evolutionary pathways (*Jochumsen et al., 2016*). Both options can be easily included in our model, but empirical data in this regard is sparse, and we expect our main results with regard to PD and PK influence on single- and multi-step resistance to be robust to such changes. The empirical data that we used to inform our simulations did also not provide explicit information about potential compensatory mutations, which arguably can influence the dynamics of resistance evolution (*Andersson and Hughes, 2010*) – although likely in a very complex manner, as recent studies suggest (*Dunai et al., 2019*). According to our results, these mutations might even be a necessary means to allow multi-step resistance patterns to arise. If they emerge fast enough to compensate for the cost of the first mutation, they would increase the selection coefficient of this mutational sub-population and thereby provide a stepping stone to high-level resistance. This might either be akin to crossing a fitness valley, if the first mutation does not provide a benefit, or it might facilitate climbing a fitness peak by making low-benefit mutations more favorable.

For many antimicrobial drugs, resistance evolution can not only arise through chromosomal mutations, but also by acquisition of resistance genes through HGT (*van Hoek et al., 2011*). Notably, our results highlight the importance of transfer rates as we find rescue of the pathogen population through HGT resistance only if the initial acquisition rate is higher than the mutation rate. HGT is not only dependent on the recipient population size but also on the donor population size, hence using typical experimentally measured conjugation rates of $10^{-11}$–$10^{-13}$ ml cell$^{-1}$ h$^{-1}$ (*Licht et al., 1999*), environmental donors have to be more abundant than $10^5$ cells ml$^{-1}$ to be faster than chromosomal mutation rates (*Rodríguez-Rojas et al., 2014*) of $10^{-6}$, which might not always be the case at bacterial infection sites (*Stecher et al., 2012*). This implies either (i) that HGT resistance is acquired after chromosomal mutations, (ii) that HGT spreads mostly at sublethal drug doses, or (iii) that acquisition rates from a pre-existent pool of HGT carriers are high. Plasmid transfer rates are likely increased at low AB doses (*Cairns et al., 2018*), but generally they are highly variable, and even though they are biased towards spread between clone-mates, there seems to be no obvious correlation between

transfer rates and genetic distance of donors and recipients (*Dimitriu et al., 2019*). Hence, determining the relative importance of resistance evolution through HGT or chromosomal mutations is difficult, but for specific drugs like AMPs, for which spread of HGT resistance from the gut microbiota seems to be low (*Kintses et al., 2019*), the risk of treatment failure is mainly shaped by the beneficial mutations available to the population.

Most of our results assumed a completely susceptible pathogen population at onset of treatment, as seen in many bacterial infections (*Balmer and Tanner, 2011*). However, the fast growth and high mutation rates can lead to significant heterogeneity in bacterial populations and we would expect this (neutral) heterogeneity to increase treatment failure, even with multi-step resistance patterns, by giving the population a 'head-start' in the accumulation of mutations. This is indeed what we see with our model when we start from a heterogeneous population, but we still find on average less than 50% treatment failure in each multi-step resistance scenario (considering various PKs and PDs) (Materials and methods, *Figure 2—figure supplement 6*, *Figure 3—figure supplement 7*).

When starting from populations that likely already contain resistance mutations, aiming for 'mitigation' (adaptive treatment) can be more effective in reducing resistance spread than trying to completely 'eradicate' the pathogen population (aggressive treatment). If multiple steps are necessary to obtain full resistance to the highest possible drug dose, we find that the threshold for choosing adaptive over aggressive treatment can be much lower than if only a single mutation were necessary (*Figure 4*). Additionally, in drug-free environments, we expect a lower frequency of resistant cells for multi-step patterns as it is less likely that neutral heterogeneity produces cells carrying all resistance mutations. Hence, the high competitive benefit is paired with a low risk for resistance evolution. Even though determination of the number and size of resistant subpopulations is very difficult in practice, this suggests that adaptive treatment is likely to be superior in containment of resistant infections for many drugs, for which multi-step patterns are the most common resistance mechanism. Further, the assumptions in our model are not specific to bacterial populations or antimicrobials, which makes them more broadly applicable to other drug treatments, like cancer therapy (*Chakrabarti and Michor, 2017*). Our results suggest a way forward to develop treatment strategies that – in addition to all other important considerations – explicitly account for the risk of drug resistance evolution.

## Materials and methods

### Key resources table

| Reagent type (species) or resource | Designation | Source or reference |
| --- | --- | --- |
| Software, algorithm | R package *adaptivetau* | Johnson, P. Tau-Leaping Stochastic Simulation. R package version 2.2–3 (2019) (*Johnson, 2* |
| Other | Previously published datasets | Melnyk A, Wong A, Kassen R. The fitness costs of antibiotic resistance mutations. (2015) (*M* |

### Literature review of costs and benefits of antimicrobial drug resistance mutations

We compounded a comprehensive set of experimental evolution studies (or reviews thereof) that measured both fitness costs (usually growth rate reductions in the absence of drugs) and benefits (usually increases in MIC) of AB or AMP resistance mutations within the same set of experiments. The studies used various bacterial species, including pathogenic isolates (see *Table 1*). From empirically measured data of sample replicates, we calculated costs as the arithmetic mean of 1-(relative fitness to wildtype) and the benefit as the geometric mean (due to the logarithmic scale of MIC/IC$_{50}$ measures) of MIC or IC$_{50}$ increase relative to the wildtype. (Note: *Chevereau et al., 2015* used IC$_{50}$ instead of MIC but our calculation of IC$_{50}$ and IC$_{90}$ – which is likely very close to MIC – in their data gave a good correlation [$R^2 = 0.45$, $p<0.001$], which indicates that the benefits obtained from IC$_{50}$ measurements are comparable to ones obtained from MIC measurements.) As fitness measure, we considered only the measurements done in the same conditions (media and temperature) that was also used for experimental evolution, even if growth was also measured in different environments. We list the conditions of the various evolution experiments, MIC and fitness measurements in

*Table 1*, with the exception of *Melnyk et al., 2015*, where we only list the eight different pathogenic strains used, as this paper synthesizes 24 different studies, grown under various conditions.

We obtained the type and average number of mutational events observed from supplemental data in most studies, but there was generally no possibility to link any individual resistance mutation with a specific cost and benefit. Hence, we divided the overall costs and benefits by the average number of observed (adaptive) mutations (i.e., mutations that were not observed in control lines), assuming that each mutation provides a similar share to the overall magnitude. As most studies have a very low number of mutational events linked to resistance, this assumption is not expected to lead to strong biases. Overall, the results from all of the studies gave only a very weak positive linear correlation between the log(benefit) and cost of a mutational event (*Figure 1B*). Mutations seem to be more likely to incur costs than benefits. This result is largely determined by the large data set from *Spohn et al., 2019*, which gives a very weak correlation between cost and benefit (*Figure 1—figure supplement 1*), similar to the data points from *Melnyk et al., 2015*. The dataset from *Spohn et al., 2019* is the only one that fulfilled our criteria and directly compared AB and AMP mutational effects, which we summarize in *Figure 1—figure supplement 1*. The calculated benefit and cost per mutation for each individual AB and AMP in the *Spohn et al., 2019* and *Melnyk et al., 2015* data is given in *Figure 1—source data 1*.

## PD model

We combined a PD model, which connects the growth of bacterial (mutant) subpopulations to antimicrobial drug concentration (*Figure 1A*; *Nielsen and Friberg, 2013*; *Andersson et al., 2020*; *Read et al., 2011*; *Clarelli et al., 2020*; *Yu et al., 2018*; *Regoes et al., 2004*), with a population model to predict the emergence of resistance mutations in individual bacterial cells.

In the population model, bacteria can grow up to a certain carrying capacity and can accumulate mutations during replication at a certain rate. In order to simulate de novo mutation emergence, we started most of our simulations from a completely susceptible population $M_0$, but we also ran simulations starting from neutral diversity (meaning that we ran the simulation for 50 hr without AB treatment and then started from the observed neutral heterogeneity) (*Figure 2—figure supplement 6*, *Figure 3—figure supplement 7*). We do not allow for reversion of resistance mutations, which has been found to be rare (*Dunai et al., 2019*) and likely does not play a role in multi-step resistance networks (over the time frame of a single treatment period). The population dynamics is captured by the following deterministic equations (which were implemented in a stochastic manner):

$$\frac{dM_i}{dt} = r \cdot (1-c)^{i-1} \cdot u \cdot M_{i-1} \cdot \left(1 - \frac{M}{K}\right) + r \cdot (1-c)^i \cdot (1-u) \cdot M_i \cdot \left(1 - \frac{M}{K}\right) - (\gamma + \gamma_i) \cdot M_i$$

$$with \ i = 0, 1, 2, \ldots$$

$$M = \sum_i M_i$$

Here, $M_i$ is the bacterial subpopulation carrying $i$ mutations, $r$ the wildtype growth rate (set to 1 in our simulations), $c$ the cost of each mutation, $u$ the mutation rate, $K$ the carrying capacity of the system, $\gamma$ the natural death rate, and $\gamma_i$ the death rate caused by drugs (which captures the PD properties of a drug and the resistance level of the bacterial population via the mutational benefit).

## The PD function

In our population model, cells die at a low intrinsic rate $\gamma$, whereas death due to antimicrobials ($\gamma_i$) is dependent on the properties of the antimicrobial applied, the benefit conferred by each mutation, and the PK profile. Specifically, $\gamma_i$ is calculated from the maximal and minimal growth rates $\psi_{max}$ and $\psi_{min}$ (note that $\psi_{min}$ can be negative in the presence of drugs, meaning that we generally consider bactericidal AB action), the (time-dependent) concentration of the drug $a$, the MIC of the mutation-free population (set to 1 in our simulations), the benefit $b_i$ conferred by each mutation, and the sensitivity of the dose–growth relationship $\kappa$ (the Hill coefficient or steepness of the curve):

$$\gamma_i = \frac{(\psi_{max} - \psi_{min}) \cdot (a/(MIC \cdot b_i))^\kappa}{(a/(MIC \cdot b_i))^\kappa - \psi_{min}/\psi_{max}}$$

$$\psi_{max} = r \cdot (1 - c)^i - \gamma$$

Considering bacteriostatic antimicrobial action can be achieved in our model by using a small $\psi_{min}$ value and incorporating antimicrobial effect into the growth, not the death term. Note that introducing antimicrobial action into the birth term here leads to density-dependent antimicrobial effects. This is not entirely unrealistic, considering persister bacteria, whose dormant state protects them from killing by ABs (*Kussell et al., 2005*). However, bacteriostatic action in itself would result in a high bacterial presence at the end of the treatment – even if bacteria are fully susceptible to the antimicrobial – as intrinsic bacterial death is very low. Hence, we incorporated an extrinsic removal rate $-\gamma_{cl} \cdot M_i$, akin to immune system clearance of inert bacterial cells, with $\gamma_{cl} = 0.1 h^{-1}$ being in a realistic range (*Roach et al., 2017*). The model for bacteriostatic drug action is then given by

$$\frac{dM_i}{dt} = \left( r \cdot (1-c)^{i-1} - \gamma_{i-1} \right) \cdot u \cdot M_{i-1} \cdot \left( 1 - \frac{M}{K} \right) + \left( r \cdot (1-c)^i - \gamma_i \right) \cdot (1-u) \cdot M_i \cdot \left( 1 - \frac{M}{K} \right) - (\gamma + \gamma_{cl}) \cdot M_i$$

*with* $i = 0, 1, 2, \ldots$

$$M = \sum_i M_i$$

## Benefits and costs of mutations

The main interest of our study is the comparison of AB resistance evolution via 'typical' single mutations with complex, multi-step processes as shown in *Jochumsen et al., 2016*. The latter are characterized by a network of mutations of small benefits in multiple genomic resistance loci that create evolutionary pathways to high-level AB resistance (*Jochumsen et al., 2016*). We model this mutation accumulation via sequential acquisition of mutations with a certain benefit and cost, that is, decreases in drug-induced death and decreases in the maximum growth rate. Benefits and costs of each mutation were taken from the positive correlation that was observed with literature values (slope = 0.0087) – except for simulations testing the dependence of our results on this relationship, where we took a steeper correlation (slope = 0.0467) (*Figure 2—figure supplement 1*).

As benefits and costs are likely to vary, we also confirmed that our results are robust with regard to drawing benefits and costs of each mutation from a normal distribution. Similarly, we ran simulations with 'peak PK', where only the first mutational benefit/cost was fixed (i.e., deciding if a single- or multi-step pattern was necessary) and the other mutations were sampled from the whole range of benefits and costs obtained from the literature, independently of each other (*Figure 2—figure supplement 5*). We ignore the possibility of positive epistasis between these mutations (which would speed up resistance evolution as fewer mutations would be required for higher levels of resistance), as well as the possibility of negative epistasis, which would limit access to some mutations and the available pathways to resistance (thereby slowing down resistance evolution as 'effective' mutation rates might be lower than we assume in our model). Both of these processes are complex and not well understood, hence by ignoring these possibilities we aim to provide a more fundamental and intuitive comparison between single- and multi-step resistance evolution.

Resistance mutation rates were generally kept the same for each simulation run (i.e., regardless of the benefit magnitude). In reality, there might be more mutations available that provide a low benefit – which are likely to be less specific and therefore have a larger genomic target size, but using higher mutation rates for mutations with lower benefits and costs – which was done proportional to the number of steps needed to obtain resistance – did not change our results noticeably (*Figure 2—figure supplement 2*).

## PK functions

In our simulations, we used three different PK functions to evaluate resistance evolution dynamics. 'Peak PK' describes the intake of a drug with a certain period $\tau$, which is absorbed instantaneously and then decays exponentially at rate $k$ (**Yu et al., 2018**):

$$a(t) = \sum_n d \cdot \left( e^{-k[t-(n-1)\tau]} \right)$$

with $n$ = 1,2,... the number of times the treatment dose $d$ is applied.

For 'constant PK', the drug concentration is independent of time and simplifies to $a = d$, whereas for 'ramp PK', the drug concentration increases linearly over a time $k2cmax$ (hence the rate of drug concentration increase is given as $d/k2cmax$) and then stays constant for the rest of the treatment period. The value for $k2cmax$ used for most simulations (48 hr) was taken from literature and describes an example of AMP production timing during a natural immune response (**Haine et al., 2008**).

## Implementation and simulation

The model was implemented in R using the package *adaptivetau* (**Johnson, 2019**) for stochastic implementations via the Gillespie algorithm. We focused on stochastic simulations as we were particularly interested in the timing and probability of the de novo rise and fixation of multiple mutations. We used the package *adaptivetau* because it allows for time-varying reaction rates, which was necessary in order to incorporate drug-concentration-dependent death rates. It also allows for deterministic simulation of a subset of rates as we did not want the AB concentration to be stochastic. For increased accuracy, we changed the epsilon parameter (which describes the tolerance of relative rate changes in step size selection) to 0.01, which we found gave the same results as exact simulations.

We calculated treatment failure probability as the frequency of runs, in which bacteria were not eradicated at the end of the treatment period (200 hr). Mutational diversity was calculated using the Shannon index, which takes into account the richness and evenness of the distribution of mutant subpopulations, and either averaging the maximum per treatment period (for most results) or the end diversity (**Figure 3—figure supplement 3**) over all simulation runs. Treatment failure probability and mutational diversity were plotted using the R function *filled.contour*, which, as far as we could ascertain, interpolates linearly between (potentially irregularly) spaced grid points. To increase the appeal of our figures, we increased the option nlevels from the default value of 20 to 50. The values for treatment failure and diversity shown in the contour plots were then averaged over the whole multi- or single-step area (colored triangles shown in **Figure 2**) in order to compare different treatment strategies.

The difference in treatment failure and mutational diversity between the two antimicrobial classes (PK profiles) was obtained by subtracting the corresponding values after every simulation of an AMP treatment from the one obtained in a simulation for an AB treatment and plotting the individual resulting differences (for 500 simulations) as well as the density via violin plots.

Model parameters other than benefit, cost, and drug dose are taken from **Yu et al., 2018** (**Supplementary file 1**). The two different antimicrobial drug classes were defined based on previous experimental and theoretical work (**Yu et al., 2018**; **Rodríguez-Rojas et al., 2014**; **Yu et al., 2016**) by using two parameter sets: for the AB class, the mutation rate was $3 * 10^{-6}$ per division, $\kappa$ was 1.5 and $\psi_{min}$ was $-5 \text{ h}^{-1}$; whereas for the AMP class, the mutation rate was $10^{-6}$, $\kappa$ was 5 and $\psi_{min}$ was $-50 \text{ h}^{-1}$.

The described code has been made available as an R package (**Source code 1**).

## Selection coefficient analysis

Selection coefficients for our PD model were calculated under the assumption that the sensitive population is very small compared to the carrying capacity, which is a good approximation to the selection pressure at the start of an infection. This means that we can neglect the logistic growth term in our calculations. As the results were very similar to assuming a population at the carrying capacity (which is an approximation for an infection that has had time to establish itself), we will focus on the

selection coefficients with a small starting population. Selection coefficients were determined through eigenvalues obtained from the Jacobi matrix given by $\frac{dM}{dt} = \begin{bmatrix} \frac{dM_0}{dt} \\ \vdots \\ \frac{dM_n}{dt} \end{bmatrix} = N(i) * \boldsymbol{M}(t)$:

$$
N(i) = \begin{pmatrix}
r \cdot (1-u) - (\gamma + \gamma_0) & 0 & \cdots & & \cdots & 0 \\
r \cdot u & r \cdot (1-c)^1 \cdot (1-u) - (\gamma + \gamma_1) & \cdots & & \cdots & \vdots \\
0 & r \cdot (1-c)^1 \cdot u & \ddots & & & \vdots \\
\vdots & \ddots & & \ddots & r \cdot (1-c)^{n-1} \cdot (1-u) - (\gamma + \gamma_{n-1}) & 0 \\
0 & \cdots & & & r \cdot (1-c)^{n-1} \cdot u & r \cdot (1-c)^n \cdot (1-u) - (\gamma + \gamma_n)
\end{pmatrix}
$$

The eigenvalues of $N(i)$ are its diagonal entries, which correspond to the net growth of each population. We calculated selection coefficients for each of the mutational subpopulations in our model as the difference in growth rates between bacteria with $i$ mutations and bacteria with $i–1$ mutations (i.e., the difference between their eigenvalues):

$$
s_i = growth(M_i) - growth(M_{i-1}) = \left( r \cdot (1-c)^i \cdot (1-u) - (\gamma + \gamma_i) \right) - \left( r \cdot (1-c)^{i-1} \cdot (1-u) - (\gamma + \gamma_{i-1}) \right)
$$

The AB concentration over time was calculated deterministically using the R package *deSolve* (*Soetaert et al., 2010*) in order to calculate the death rates due to antimicrobial treatment.

The difference between the parameter sets for the two antimicrobial classes used here lies in the higher mutation rate $u$, lower $\kappa$, and higher $\psi_{min}$ for AB treatments (*Yu et al., 2018*). Hence, we investigated the importance of the two PD parameters $\kappa$ and $\psi_{min}$ by calculating the selection coefficients using the AMP parameter set and swapping either $\kappa$ or $\psi_{min}$ with that of the AB parameter set.

More generally, we can consider the Jacobi matrix for the resistant populations invading at the mutant-free equilibrium:

$$
N(i) = \begin{pmatrix}
r \cdot (1-c)^1 \cdot (1-u) \cdot \left(1 - \frac{M_0^*}{K}\right) - (\gamma + \gamma_1) & 0 & \cdots & & \cdots & 0 \\
r \cdot (1-c)^1 \cdot u \cdot \left(1 - \frac{M_0^*}{K}\right) & r \cdot (1-c)^2 \cdot (1-u) \cdot \left(1 - \frac{M_0^*}{K}\right) - (\gamma + \gamma_2) & 0 & & \cdots & \vdots \\
0 & \cdots & \ddots & & \ddots & \vdots \\
\vdots & \ddots & & r \cdot (1-c)^{n-1} \cdot (1-u) \cdot \left(1 - \frac{M_0^*}{K}\right) - (\gamma + \gamma_{n-1}) & & 0 \\
0 & \cdots & & r \cdot (1-c)^n \cdot u \cdot \left(1 - \frac{M_0^*}{K}\right) & r \cdot (1-c)^n \cdot (1-u) \cdot \left(1 - \frac{M^*}{K}\right) - (\gamma + \gamma_n)
\end{pmatrix}
$$

The criteria for invasion of a mutant into the susceptible population is then that the eigenvalue of the mutant has to be bigger than zero, that is,

$$
\lambda_n = r \cdot (1-c)^n \cdot (1-u) \cdot \left(1 - \frac{M_0^*}{K}\right) - (\gamma + \gamma_n) > 0
$$

Inserting the mutant-free equilibrium $M_0^* = K \cdot \left(1 - \frac{\gamma + \gamma_0}{r \cdot (1-u)}\right)$ yields

$$
\gamma + \gamma_0 > \frac{\gamma + \gamma_n}{(1-c)^n}
$$

which means that bacterial cells with n mutations can invade if the death rate of the sensitive strain is higher than the death rate of the mutant normalized by the cost of the mutation(s).

## Horizontal gene transfer

We added HGT to the model by allowing for an additional resistance gene (with benefit $b_p$ and cost $c_p$) to be acquired, which gives resistance in a single step. Hence, the benefit $b_p$ and the corresponding cost $c_p$ were adjusted with respect to the drug dose applied by using a benefit that would increase the MIC 20% above the applied drug dose and calculating the cost accordingly through the linear correlation obtained from *Table 1*. We assume that the bacterial population under investigation has not yet acquired the HGT element, and initial transfer has to come from the environment, that is, initial conditions were the same as for simulations without HGT and $M_p(0) = 0$. This gene can

be acquired at a low rate $\alpha$ from the environment or at a density-dependent rate $\beta$, which we assumed to be on the same order of magnitude as the mutation rate (*Bakkeren et al., 2019*).

The horizontally transferred gene can be acquired by sensitive or mutant bacterial populations, and cells containing HGT resistance can still acquire further mutations (but not further HGT resistance). Hence, we assume that HGT resistance is, for example, acquired via a specific resistance gene on a plasmid (typically a plasmid can only be acquired once per cell) and that the resistance gene from this plasmid (e.g., using enzymatic drug inactivation) acts through a different mechanism than resistance by chromosomal mutation (e.g., modification of the drug target) (*van Hoek et al., 2011*).

The equations were modified as follows:

$$\frac{dM_i}{dt} = r \cdot (1-c)^{i-1} \cdot u \cdot M_{i-1} \cdot \left(1 - \frac{M}{K}\right) + r \cdot (1-c)^i \cdot (1-u) \cdot M_i \cdot \left(1 - \frac{M}{K}\right) - (\gamma + \gamma_i) \cdot M_i - (\alpha + \beta \cdot M_p) \cdot M_i$$

$$\frac{dM_{pi}}{dt} = r \cdot \left((1-c)^{i-1} * c_p\right) \cdot u \cdot M_{pi-1} \cdot \left(1 - \frac{M}{K}\right) + r \cdot \left((1-c)^i * c_p\right) \cdot (1-u) \cdot M_{pi} \cdot \left(1 - \frac{M}{K}\right) - (\gamma + \gamma_{pi}) \cdot M_{pi} + (\alpha + \beta \cdot M_p) \cdot M_i$$

$$M = \sum_i M_i + \sum_i M_{pi}$$

$$M_p = \sum_i M_{pi}$$

Here, $M_{pi}$ is the bacterial subpopulation carrying the HGT gene and $i$ mutations, $M_p$ the total number of HGT subpopulations, and $M$ the total number of all bacterial populations. Relative population frequencies were calculated at the end of the treatment period by dividing the cell number of each subpopulation through the whole population size.

## Adaptive treatment

In adaptive treatment, the goal is not to eradicate the bacterial population entirely but to adjust the treatment dose continuously in order to keep the pathogen level below a certain upper limit. *Hansen et al., 2017* calculated the threshold of resistant cells that are necessary at the beginning of the treatment for adaptive treatment to outperform aggressive treatment (i.e., giving the full dose right away), which is based on the idea that sensitive cells provide a risk for becoming resistant through mutation and a benefit through growth competition with the resistant cells at the same time. *Hansen et al., 2017* only considered one mutation to resistance, which means that their risk subpopulation and competitor subpopulation was the same. If we consider however sequential mutational steps, then the risk population only consists of the subpopulation one mutation away from full resistance (which will be the mth mutation), whereas the competitor population for the fully resistant strain contains all (partially) sensitive bacteria (i.e., including mutant strains, which are not fully resistant to the highest possible treatment dose). Therefore, the threshold of resistant bacteria is given by (compare to [4] in *Hansen et al., 2017*):

$$\text{Competitive benefit of (partially) sensitives} = \text{Mutational risk from } M_{m-1}$$

$$r \cdot (1-c)^m \cdot M_m \cdot \delta \cdot (P_{max} - M_m) = u \cdot r \cdot (1-c)^{m-1} \cdot (1 - \delta P_{max}) \cdot M_{m-1}$$

$$M_m = \frac{u \cdot (1 - \delta P_{max}) \cdot M_{m-1}}{(1-c) \cdot \delta \cdot \sum_{j=0}^{m-1} M_j} \quad \text{with} (P_{max} - M_m) = \sum_{j=0}^{m-1} M_j$$

Here, $\delta$ describes the strength of competition and $P_{max}$ the upper limit of acceptable pathogen burden. Note that here it is assumed that all pathogens (regardless of drug sensitivity) contribute equally to competition (*Hansen et al., 2017*). This leads to a quadratic equation for the subpopulation with m mutations, $M_m$, which we used to calculate how the resistant population threshold for

adaptive treatment (i.e., the initial density $M_{m0}$ above which adaptive treatment is more favorable) differs between multi- and single-step resistance patterns (*Figure 4*).

We implemented adaptive treatment in our model by setting a defined upper bound of acceptable pathogen cells, which was equal to the starting density in these simulations (*Hansen et al., 2017*) (i.e., assuming that the bacterial infection already progressed to a level at which treatment becomes necessary). We used a relatively low acceptable burden of $10^5$ CFU, which is supported by bacterial loads in, for example, urinary tract infections (*Schmiemann et al., 2010*). Note that defining an acceptable limit of pathogen burden in clinical settings is far more intricate as a patient's individual biology will play a significant role and is beyond the scope of this paper. We adjusted the treatment dose in order to keep the pathogen load at or below this threshold value but the subpopulations of at least partially sensitive cells as big as possible (*Figure 4—figure supplement 1*): specifically, we increased the treatment dose to the MIC of the highest resistant subpopulation when its frequency exceeded 1% of the total population and the total pathogen load was higher than our set acceptable burden – until the maximum dose set for a specific treatment simulation was reached; after which the maximum dose was applied continuously. The (partially) sensitive cells serve as competitors for the resistant strain that carries a mutational growth cost and can be outcompeted at low drug doses (*Hansen et al., 2017*). At the same time, subpopulations that are one step away from the resistant population provide a risk population as they are likely to gain resistance.

For simulations of adaptive and aggressive treatment, we started from a population with neutral heterogeneity, meaning that we calculated the steady-state number of cells with a specific number of mutations given a certain cost (and benefit) in the absence of drug selection. As we want to compare the time difference to treatment failure between adaptive and aggressive treatment for single- and multi-step patterns, we initially add to this 'neutral population' the predicted number of resistant cells necessary to make adaptive treatment superior to aggressive treatment. The drug dose in adaptive treatments was then adjusted to keep the number of pathogens below the acceptable burden as described above. The time of treatment failure was determined as the time where the total pathogen population crossed $10^8$ CFUs. We compared adaptive and aggressive treatment by dividing the time to treatment failure obtained from the adaptive strategy by the one obtained with the aggressive strategy, yielding the fold difference in treatment success duration.

## Acknowledgements

We thank D Baeder, S Bonhoeffer, S Lehtinen, and H Alexander for useful discussions and comments on the manuscript. This work was supported by a grant from the Volkswagen Foundation (grant no. 96517).

## Additional information

### Funding

| Funder | Grant reference number | Author |
| --- | --- | --- |
| Volkswagen Foundation | 96517 | Claudia Igler<br>Jens Rolff<br>Roland Regoes |

The funders had no role in study design, data collection and interpretation, or the decision to submit the work for publication.

### Author contributions

Claudia Igler, Conceptualization, Data curation, Software, Formal analysis, Investigation, Visualization, Methodology, Writing - original draft, Writing - review and editing; Jens Rolff, Conceptualization, Writing - review and editing; Roland Regoes, Conceptualization, Supervision, Funding acquisition, Methodology, Writing - review and editing

## Author ORCIDs
Claudia Igler (iD) https://orcid.org/0000-0001-7777-546X
Jens Rolff (iD) http://orcid.org/0000-0002-1529-5409
Roland Regoes (iD) https://orcid.org/0000-0001-8319-5293

## Decision letter and Author response
Decision letter https://doi.org/10.7554/eLife.64116.sa1
Author response https://doi.org/10.7554/eLife.64116.sa2

## Additional files

### Supplementary files
• Source code 1. R package containing the pharmacokinetic and pharmacodynamic model for multi-step resistance evolution. R code used for the simulations shown in *Figures 2–4* and figure supplements (with the exception of *Figure 1—figure supplement 1*). Documentation of each function is included in the package.

• Supplementary file 1. Model parameters. Parameter values and units used in the pharmacodynamic model for either antibiotic or antimicrobial peptide simulations are shown.

• Transparent reporting form

### Data availability
All data and code generated or analysed during this study are included in the manuscript and supporting files. Source code has been provided for Figures 2-4, as well as Figure 1—figure supplement 2, Figure 2—figure supplements 1–6, Figure 3—figure supplements 1–7, and Figure 4—figure supplements 1–2 in the form of an R package. Source data has been provided for Table 1, Figure 1B and Figure 1—figure supplement 1.

The following previously published dataset was used:

| Author(s) | Year | Dataset title | Dataset URL | Database and Identifier |
|---|---|---|---|---|
| Melnyk A, Wong A, Kassen R | 2015 | The fitness costs of antibiotic resistance mutations | https://doi.org/10.5061/dryad.5rc47 | Dryad Digital Repository, 10.5061/dryad.5rc47 |

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
