## [Decision Letter]

**Acceptance summary:**

Previously mathematical models have explored how different antibiotic treatment strategies might affect the emergence of antibiotic resistance and subsequent treatment failure, but these have all assumed that resistance arises in a single mutational step. In practice, two or more mutational steps are often needed before clinically significant resistance arises. This paper address this gap, and explores through a series of computer simulations the potential impact of multiple mutational steps on the likelihood of resistance emergence, and the impact of different treatment strategies on such resistance emerging and reveals unexpected complexity in predicting which treatment strategies will minimize resistance evolution.

**Decision letter after peer review:**

Thank you for submitting your article "Multi-step vs. single-step resistance evolution under different drugs, pharmacokinetics and treatment regimens" for consideration by *eLife*. Your article has been reviewed by 2 peer reviewers, and the evaluation has been overseen by a Reviewing Editor and Patricia Wittkopp as the Senior Editor. The reviewers have opted to remain anonymous.

The reviewers have discussed the reviews with one another and the Reviewing Editor has drafted this decision to help you prepare a revised submission.

Summary:

The emergence of antibiotic resistance through mutations that arise during treatment is of clinical importance for some bacterial species and certain antibiotics. Previously mathematical models have explored how different antibiotic treatment strategies might affect the emergence of such resistance and subsequent treatment failure. However, these models have all assumed that resistance arises in a single mutational step. In practice, two or more mutational steps are often needed before clinically significant resistance arises. This paper address this gap, and explores through a series of computer simulations the potential impact of multiple mutational steps on the likelihood of resistance emergence, and the impact of different treatment strategies on such resistance emerging. The paper also includes a literature review of the costs and benefits of mutations conferring resistance.

Essential revisions:

There are a number of areas where there is a need to clarify aspects of the work, provide additional information, or add to the discussion. Full reviews are appended and all points should be addressed with the exception that, following the discussion, the reviewers and editor agreed there was not a need to reduce the amount of material in the paper.

Specific points that need addressing (not all of which are mentioned in the appended reviews) are:

1. There is a need to improve the framing of the work. There is a danger that the work gives the impression that emergence of resistance through mutations is the primary driver of clinically important antimicrobial resistance. As reference 22 makes clear, this is not the case, though this route to resistance is important for certain bacteria (such as TB) and certain antimicrobial agents. The introduction should highlight this point, and give examples of organisms and antibiotics for which this work is relevant. It also needs to be clearer that the treatment of HGT is only relevant for the limited range of bug-drug combinations where such emergence of resistance during treatment is important.

2. There is also a danger that readers may get the mistaken impression that emergence of resistance is the only consideration when considering dosing. This is clearly not the case, and the best treatment strategies for minimising risk of treatment failure due to resistance are not necessarily the best strategies for patient outcomes overall. Again, the authors should make this clear in the introduction, and make it clear that any findings in this paper should not be used to directly influence patient treatment decisions, and highlight how they do envisage these findings being used. This is important, as modelling studies are often misinterpreted and taken out of context, and without such explicit caveats there is a danger that they are used to support dosing decisions that might cause real harm to patients.

3. Table 1 needs more detail about the nature of the organisms and the environments they are growing in.

4. Clarification of rationale for choice of initial conditions in simulations is needed

5. Need to explain more clearly why diversity is being looked at and why the measure used to measure it is appropriate.

6. More detailed discussion that makes it clearer how the modelling has added to our understanding is needed.

7. There is a need to highlight that antimicrobials are assumed to affect the death rate of bacteria and not their birth rate (which has implications for the type of antibiotics this modelling is relevant for). Can the discussion consider what changes would be expected if this assumption were to be relaxed?

8. Methods need to be described with greater precision – which methods were used when, what approximations were made, and why methods were chosen.

9. Figure 2 needs more text to help readers to interpret it.

*Reviewer #1:*

This manuscript considers how the details of resistance evolution impact genetic diversity and the probability of treatment failure. Specifically it uses an in silico analysis to compare resistance evolving from single step mutation versus multi-step mutation. Some of the basic patterns concerning costs and benefits of mutation are motivated by a literature search. This study investigates an important and interesting question and is quite ambitious in its undertaking. In addition to the main topic described above it also considers the additional complications of PK/PD dynamics (by considering three different dosing profiles and two types of antimicrobial), horizontal gene transfer and adaptive therapy. My sense is that the manuscript would be much stronger if it covered fewer topics but covered these topics more comprehensively (for example, a more detailed discussion on diversity would be more informative to the reader). In general it was very hard for me to understand how heavily the results depended on the model structure/assumptions.

1. Table 1 is very nice and contains a lot of interesting information. The cost-benefit relationship of mutations is probably heavily influenced by (i) the environment that the mutations evolved in and (ii) the environment the costs and benefits are measured in. Can you indicate in the table when these environments differ from each other? Also, do any of the strains originate from clinical isolates? I wasn't sure if the data in the Table reflected trends I would expect to see in a patient.

2. Simulations assumed that the initial population was completely susceptible (except for the adaptive therapy analysis). I don't understand why this was done. Why wouldn't you start the simulations with a single sensitive bacteria and then use the dynamical equations to grow the population to the "starting population size". That would give you an initial population with a diversity that reflects the assumed dynamics.

3. Can you clarify the reason you are looking at diversity and why the measure you use is appropriate to investigate the questions you have? I was confused about the decision to average diversity over 200 hours, especially since some populations must go extinct well before 200 hours? For populations that don't go extinct I'd be interested to know how the process of treatment modified the diversity (i.e., what is the composition of the population at the end of 200 hours?) In general I found the discussion around diversity confusing. (I suspect the conclusion on L313-314 is heavily influenced by the way diversity is measured in this study.)

4. I would find the paper much more impactful if the main findings listed in the abstract (L21-25) were discussed in more detail in the discussion. For example, the first result that "resistance evolution is substantially limited if two or more mutations are required": before the modeling analysis, under what situations would you not expect this to be the case? How did the modeling help rule these situations out? Etc.

*Reviewer #2:*

In their manuscript titled "Multi-step vs. single-step resistance evolution under different drugs, pharmacokinetics and treatment regimens", the authors present a very comprehensive study of antimicrobial resistance evolution, based on stochastic simulations of microbial population growth incorporating pharmacokinetic and pharmacodynamic models of various antimicrobial treatments, as well as experimentally-motivated properties of the mutational pathways to resistance. They compare resistance evolution via a single large-effect mutation to resistance evolution via a sequence of mutations, antibiotics to antimicrobial peptides, and different treatment regimens, including adaptive treatment. They mainly focus on de novo mutations but also include horizontal gene transfer. I found this paper extremely interesting and well-written, incorporating many crucial aspects of resistance evolution in an attractive unified framework, and providing multiple important and nontrivial conclusions on how to optimize antimicrobial treatments. It is also very timely, given the pressing public health issue raised by antimicrobial resistance.

In the model developed here (line 443), it is assumed that the antimicrobial affects the death rate of bacteria and not their birth rate, since its effect is in the death rate γ_i_ caused by drugs. In other words, the focus is on bactericidal drugs. In practice, some antibiotics are bacteriostatic, and many possess combined bacteriostatic and bactericidal effects. It would be good to point out that this assumption is made, and also to discuss what changes would be expected if bacteriostatic or mixed effects were considered.

The description of the model details in "Implementation and simulation" is rather brief. In particular, it is mentioned that "deterministic analysis" and "stochastic implementations via the Gillespie algorithm" were both carried out. It would be important to specify where each of these approaches was employed. More generally, it would also be helpful to discuss why including stochastic effects is important in the analysis, and for which specific points it is crucial. From a technical point of view, it would also be valuable to explain what approximations are made in the specific tau-leaping R package employed for the Gillespie simulations, and how time-varying reaction rate constants (here, death rates impacted by concentration changes) are dealt with in this Gillespie algorithm.

The presentation of Figure 2 (and Figures S2 to S6 and Figure S13) is a bit unusual, with two triangular regions corresponding to single-step and multiple-step mutational pathways. I think that adding a few extra words about this would help the reader. In particular, it would be good to explain why one transitions from single-step to multiple-step pathways on the diagonal – it makes sense, but it is not entirely straightforward to figure this out when first seeing such plots. In addition, it would be good to clarify how (discrete) simulation results are interpolated to obtain these plots.

Apart from my three main remarks and suggestions above, here are some additional more detailed points.

It would also be good to discuss some assumptions in more detail:

l.276 : "We assumed that resistance through mutation or HGT can be acquired independently of each other and that their effects are multiplicative": it's not obvious to me why this should be the case, as one might imagine that a mutation giving full resistance can also be acquired by HGT, and then the effect of having 1 or 2 copies would depend on the resistance mechanism.

l. 409: "the benefit as the geometric mean of MIC or IC 50 increase relative to the wildtype": Please explain this choice.

l.439: "we don't allow for loss of mutations": Please justify this.

l.462: "Mutation rates were generally kept constant but considering higher mutation rates for mutations with lower benefits and costs did not change our results noticeably (Figure S4)": It would be good to explain why higher mutation rates may be associated to mutations with lower benefits and costs. Is it just that there should be more of these?

l. 550: "Hence, the benefit b_p_ and the corresponding cost c_p_ were adjusted according to the (maximum) drug dose applied.": please explain how this was done.

l. 560: Please explain what initial conditions are taken, especially for the number Mp of bact that can transfer the resistance by HGT.

l.597: "setting an upper bound of acceptable pathogen cells": please explain how it should be chosen.

l. 597: "We implemented adaptive treatment in our model by setting an upper bound of acceptable pathogen cells, and adjusting the treatment dose in order to keep the pathogen load at or below this threshold but the subpopulations of at least partially sensitive cells as big as possible (Figure S9)": please explain how this was done.

---

## [Author Response]

Essential revisions:There are a number of areas where there is a need to clarify aspects of the work, provide additional information, or add to the discussion. Full reviews are appended and all points should be addressed with the exception that, following the discussion, the reviewers and editor agreed there was not a need to reduce the amount of material in the paper.Specific points that need addressing (not all of which are mentioned in the appended reviews) are:1. There is a need to improve the framing of the work. There is a danger that the work gives the impression that emergence of resistance through mutations is the primary driver of clinically important antimicrobial resistance. As reference 22 makes clear, this is not the case, though this route to resistance is important for certain bacteria (such as TB) and certain antimicrobial agents. The introduction should highlight this point, and give examples of organisms and antibiotics for which this work is relevant. It also needs to be clearer that the treatment of HGT is only relevant for the limited range of bug-drug combinations where such emergence of resistance during treatment is important.

We agree that we did not make it clear enough for which types of infections and treatments our work will be relevant. We included a paragraph discussing clinical examples in the introduction (L55-63).

2. There is also a danger that readers may get the mistaken impression that emergence of resistance is the only consideration when considering dosing. This is clearly not the case, and the best treatment strategies for minimising risk of treatment failure due to resistance are not necessarily the best strategies for patient outcomes overall. Again, the authors should make this clear in the introduction, and make it clear that any findings in this paper should not be used to directly influence patient treatment decisions, and highlight how they do envisage these findings being used. This is important, as modelling studies are often misinterpreted and taken out of context, and without such explicit caveats there is a danger that they are used to support dosing decisions that might cause real harm to patients.

We thank the reviewers for pointing this out as we did not intend this work to be a treatment guideline, but to provide a more comprehensive view of antimicrobial resistance evolution via mutations. This has been addressed at several points in the introduction and discussion now (e.g. L29-33, L442ff).

3. Table 1 needs more detail about the nature of the organisms and the environments they are growing in.

This information was added to Table 1 by adding the column “Organism and evolution environment”, and extending the description of benefit and fitness measurements – with the one exception of the Melnyk, Wong and Kassen, 2015 paper (which we explain in the Methods section L449-453), as they themselves review and synthesize 24 different experimental studies, performed under various conditions and using different species (where we only list the species used but not all of the conditions).

4. Clarification of rationale for choice of initial conditions in simulations is needed

We explain now in more detail what our initial conditions in each of the simulations, which was added in the main text (L146f) and in more detail in the Methods section (L489-496). Further, we ran simulations with different initial conditions (starting from neutral heterogeneity), which is discussed in L418-426 and shown in Figure 2—figure supplement 6, Figure 3—figure supplement 7.

5. Need to explain more clearly why diversity is being looked at and why the measure used to measure it is appropriate.

We agree that this measure is not self-explanatory and needed more discussion. This was done in L158-166 and the Methods section (L588-592). We also compare our measure of diversity to a potentially more intuitive one, by calculating diversity only at treatment endpoints, which reduces diversity in comparison to our measure and show the results in Figure 3—figure supplement 3. We also highlight that we are interested in the maximal diversity over the treatment period because it provides information about the usefulness of increasing drug doses during the treatment, or if higher-level resistance mutations are likely to exist already in the population and will then be selected for (L326-333).

6. More detailed discussion that makes it clearer how the modelling has added to our understanding is needed.

We added more detail to the discussion explaining what the expectations prior to this modelling analysis would be and how the outcomes either contradict or confirm them. For example, while it is intuitive that the necessity for more mutations will limit resistance evolution, it is not a priori clear how strong that limitation is and how that plays out for different drug PKs, like ramp PKs, where acquisition of mutations should be facilitated. These points were added to the discussion in various places, for example L291-294, L363-369, L440-444.

7. There is a need to highlight that antimicrobials are assumed to affect the death rate of bacteria and not their birth rate (which has implications for the type of antibiotics this modelling is relevant for). Can the discussion consider what changes would be expected if this assumption were to be relaxed?

We thank the reviewers for drawing our attention to this point. We explain now in the main part and the Methods section more clearly that this assumption was made. But to get more insight into the implications of this assumption, we also ran simulations of bacteriostatic drug action and included its discussion in the Results section (L223-232), the discussion (L342ff) and the Methods section (L516-529). Even though we would have expected a slow-down of resistance evolution overall, due to the coupling of mutation emergence to replication (which is slowed down by bacteriostatic drugs), we found that this was not true for peak PKs, which showed an even higher probability of treatment failure.

8. Methods need to be described with greater precision – which methods were used when, what approximations were made, and why methods were chosen.

This was added to some extent in the main text but in more detail in every section of the Methods section describing a set of simulation runs.

9. Figure 2 needs more text to help readers to interpret it.

We added more details on the presentation and interpretation of our results to the legend of Figure 2:

“The diagonal line shows where the benefit per mutation is exactly equal to the given drug dose and separates single-step (ss, lower orange triangle), where one mutation gives a benefit higher than the applied dose, from multi-step (ms, upper blue triangle) resistance, where more than one mutation is needed for the accumulated benefit to match the drug dose.”

Reviewer #1:This manuscript considers how the details of resistance evolution impact genetic diversity and the probability of treatment failure. Specifically it uses an in silico analysis to compare resistance evolving from single step mutation versus multi-step mutation. Some of the basic patterns concerning costs and benefits of mutation are motivated by a literature search. This study investigates an important and interesting question and is quite ambitious in its undertaking. In addition to the main topic described above it also considers the additional complications of PK/PD dynamics (by considering three different dosing profiles and two types of antimicrobial), horizontal gene transfer and adaptive therapy. My sense is that the manuscript would be much stronger if it covered fewer topics but covered these topics more comprehensively (for example, a more detailed discussion on diversity would be more informative to the reader). In general it was very hard for me to understand how heavily the results depended on the model structure/assumptions.1. Table 1 is very nice and contains a lot of interesting information. The cost-benefit relationship of mutations is probably heavily influenced by (i) the environment that the mutations evolved in and (ii) the environment the costs and benefits are measured in. Can you indicate in the table when these environments differ from each other? Also, do any of the strains originate from clinical isolates? I wasn't sure if the data in the Table reflected trends I would expect to see in a patient.

We added the column “Organism and evolution environment” to Table 1 and gave more detailed descriptions in the columns for benefit and fitness measurements. However, for Melnyk, Wong and Kassen, 2015, we only state overall the various species used in the experiments (all of which were pathogenic), as they themselves review and synthesize 24 different experimental studies, which were performed under various conditions (we also explain this in the Methods section L449-453). (See also reply to Editor)

2. Simulations assumed that the initial population was completely susceptible (except for the adaptive therapy analysis). I don't understand why this was done. Why wouldn't you start the simulations with a single sensitive bacteria and then use the dynamical equations to grow the population to the "starting population size". That would give you an initial population with a diversity that reflects the assumed dynamics.

The reviewer brings up a very good point. While we were generally interested in de novo acquisition of resistance mutations, we ran further simulations starting from a neutral population structure, which gave very similar results but an (expectedly) higher treatment failure probability overall, and added this discussion in L418-426 and Figure 2—figure supplement 6, Figure 3—figure supplement 7. (See also reply to Editor)

3. Can you clarify the reason you are looking at diversity and why the measure you use is appropriate to investigate the questions you have? I was confused about the decision to average diversity over 200 hours, especially since some populations must go extinct well before 200 hours? For populations that don't go extinct I'd be interested to know how the process of treatment modified the diversity (i.e., what is the composition of the population at the end of 200 hours?) In general I found the discussion around diversity confusing. (I suspect the conclusion on L313-314 is heavily influenced by the way diversity is measured in this study.)

We agree with the reviewer that our measure of diversity is not immediately intuitive and needs more explanation and discussion. We added this in the main text in L158-166, L326-333 and the Methods section (L588-592). Further, we compare our measure with diversity obtained at the end of treatment, which indeed is much lower than the diversity that can be attained during the treatment, and added this Figure to the SI (Figure 3—figure supplement 3). (See also reply to Editor).

4. I would find the paper much more impactful if the main findings listed in the abstract (L21-25) were discussed in more detail in the discussion. For example, the first result that "resistance evolution is substantially limited if two or more mutations are required": before the modeling analysis, under what situations would you not expect this to be the case? How did the modeling help rule these situations out? Etc.

We discuss now in more detail when we would expect multi-step resistance to deter resistance evolution or when we would expect it to be facilitated and what our model shows in that regard (e.g. L291-294, L363-369, L440-444). (See also reply to Editor).

Reviewer #2:[…] In the model developed here (line 443), it is assumed that the antimicrobial affects the death rate of bacteria and not their birth rate, since its effect is in the death rate γ_i_ caused by drugs. In other words, the focus is on bactericidal drugs. In practice, some antibiotics are bacteriostatic, and many possess combined bacteriostatic and bactericidal effects. It would be good to point out that this assumption is made, and also to discuss what changes would be expected if bacteriostatic or mixed effects were considered.

We thank the reviewer for bringing our attention to this important distinction. We now clarify that we are looking at bactericidal drug effects in the main text (L148) and the Methods section (L542f). Further, we simulated bacteriostatic drug action (together with ‘immune system’ removal) and included a discussion on the comparison between bactericidal and bacteriostatic effects in the Results section (L223-232), the discussion (L342ff) and the Methods section (L516-529). (See also reply to Editor)

The description of the model details in "Implementation and simulation" is rather brief. In particular, it is mentioned that "deterministic analysis" and "stochastic implementations via the Gillespie algorithm" were both carried out. It would be important to specify where each of these approaches was employed. More generally, it would also be helpful to discuss why including stochastic effects is important in the analysis, and for which specific points it is crucial. From a technical point of view, it would also be valuable to explain what approximations are made in the specific tau-leaping R package employed for the Gillespie simulations, and how time-varying reaction rate constants (here, death rates impacted by concentration changes) are dealt with in this Gillespie algorithm.

We agree that this was written in a confusing and insufficient manner. In fact, deterministic simulations were only used for selection coefficient analysis, all other simulations were done stochastically. We explain the rationale behind using stochastic simulations (‘to investigate the timing and probability of the de novo resistance’) and give more details on the package used in the Methods section (L578-586).

The presentation of Figure 2 (and Figures S2 to S6 and Figure S13) is a bit unusual, with two triangular regions corresponding to single-step and multiple-step mutational pathways. I think that adding a few extra words about this would help the reader. In particular, it would be good to explain why one transitions from single-step to multiple-step pathways on the diagonal – it makes sense, but it is not entirely straightforward to figure this out when first seeing such plots. In addition, it would be good to clarify how (discrete) simulation results are interpolated to obtain these plots.

We thank the reviewer for the suggestions and we included more details on the presentation as well as the intuition behind the plots in the figure legend:

“The diagonal line shows where the benefit per mutation is exactly equal to the given drug dose and separates single-step (ss, lower orange triangle), where one mutation gives a benefit higher than the applied dose, from multi-step (ms, upper blue triangle) resistance, where more than one mutation is needed for the accumulated benefit to match the drug dose.”

Further, we added a sentence explaining that the R function filled.contour seems to interpolate discrete grid points linearly to obtain contour plots as shown (L592-595).

Apart from my three main remarks and suggestions above, here are some additional more detailed points.It would also be good to discuss some assumptions in more detail:l.276 : "We assumed that resistance through mutation or HGT can be acquired independently of each other and that their effects are multiplicative": it's not obvious to me why this should be the case, as one might imagine that a mutation giving full resistance can also be acquired by HGT, and then the effect of having 1 or 2 copies would depend on the resistance mechanism.

We explain now in the main text (L272ff) as well as in the Methods section (L664-668) that this was done because we are assuming that HGT resistance arises through a specific gene, while mutational resistance occurs due to changes in drug targets. Hence, we assume that these pathways are independent of each other and that HGT resistance is already ‘maximal’ and cannot be acquired twice.

l. 409: "the benefit as the geometric mean of MIC or IC 50 increase relative to the wildtype": Please explain this choice.

We added that this was done because MIC and IC50 are both measures with logarithmic scales (L455f).

l.439: "we don't allow for loss of mutations": Please justify this.

We added a paragraph explaining that we chose to specifically model sequential mutations in order to mimic mutational resistance networks as found in Jochumsen et al., 2016, but that we ignore other effects like loss of resistance mutations, which was found to be rare and is likely playing a minor role in evolutionary trajectories in such networks (L492ff, L532-538).

l.462: "Mutation rates were generally kept constant but considering higher mutation rates for mutations with lower benefits and costs did not change our results noticeably (Figure S4)": It would be good to explain why higher mutation rates may be associated to mutations with lower benefits and costs. Is it just that there should be more of these?

Yes, we would assume that mutations that provide only a small benefit would be more numerous and therefore provide a larger mutational target. We clarified this in L556-560.

l. 550: "Hence, the benefit b_p_ and the corresponding cost c_p_ were adjusted according to the (maximum) drug dose applied.": please explain how this was done.

We added that the benefit was chosen to provide an MIC 20% above the given drug dose and that cost was calculated via the correlation with benefit (L653-656).

l. 560: Please explain what initial conditions are taken, especially for the number Mp of bact that can transfer the resistance by HGT.

We clarified that we are starting from a population that has to acquire HGT resistance first from the environment, i.e. Mp=0, in the main text (L268ff) and in the Methods (L656-661).

l.597: "setting an upper bound of acceptable pathogen cells": please explain how it should be chosen.

We added the threshold value that we were using in the simulations and an explanation that this problem is much more intricate in clinical settings (L712-718).

l. 597: "We implemented adaptive treatment in our model by setting an upper bound of acceptable pathogen cells, and adjusting the treatment dose in order to keep the pathogen load at or below this threshold but the subpopulations of at least partially sensitive cells as big as possible (Figure S9)": please explain how this was done.

We added an detailed explanation to the Methods section: ‘we increased the treatment dose to the MIC of the highest resistant subpopulation when its frequency exceeded 1% of the total population and the total pathogen load was higher than our set acceptable burden – until the maximum dose set for a specific treatment simulation was reached; after which the maximum dose was applied continuously’ (L718-724).